# Communication Complexity of Distributed Convex Learning and Optimization

**Yossi Arjevani**
Weizmann Institute of Science
Rehovot 7610001, Israel
yossi.arjevani@weizmann.ac.il

**Ohad Shamir**
Weizmann Institute of Science
Rehovot 7610001, Israel
ohad.shamir@weizmann.ac.il

## Abstract

We study the fundamental limits to communication-efficient distributed methods for convex learning and optimization, under different assumptions on the information available to individual machines, and the types of functions considered. We identify cases where existing algorithms are already worst-case optimal, as well as cases where room for further improvement is still possible. Among other things, our results indicate that without similarity between the local objective functions (due to statistical data similarity or otherwise) many communication rounds may be required, even if the machines have unbounded computational power.

## 1 Introduction

We consider the problem of distributed convex learning and optimization, where a set of $m$ machines, each with access to a different local convex function $F_i : \mathbb{R}^d \mapsto \mathbb{R}$ and a convex domain $\mathcal{W} \subseteq \mathbb{R}^d$, attempt to solve the optimization problem

$$\min_{\mathbf{w} \in \mathcal{W}} F(\mathbf{w}) \quad \text{where} \quad F(\mathbf{w}) = \frac{1}{m} \sum_{i=1}^{m} F_i(\mathbf{w}). \tag{1}$$

A prominent application is empirical risk minimization, where the goal is to minimize the average loss over some dataset, where each machine has access to a different subset of the data. Letting $\{\mathbf{z}_1, \ldots, \mathbf{z}_N\}$ be the dataset composed of $N$ examples, and assuming the loss function $\ell(\mathbf{w}, \mathbf{z})$ is convex in $\mathbf{w}$, then the empirical risk minimization problem $\min_{\mathbf{w} \in \mathcal{W}} \frac{1}{N} \sum_{i=1}^{N} \ell(\mathbf{w}, \mathbf{z}_i)$ can be written as in Eq. (1), where $F_i(\mathbf{w})$ is the average loss over machine $i$'s examples.

The main challenge in solving such problems is that communication between the different machines is usually slow and constrained, at least compared to the speed of local processing. On the other hand, the datasets involved in distributed learning are usually large and high-dimensional. Therefore, machines cannot simply communicate their entire data to each other, and the question is how well can we solve problems such as Eq. (1) using as little communication as possible.

As datasets continue to increase in size, and parallel computing platforms becoming more and more common (from multiple cores on a single CPU to large-scale and geographically distributed computing grids), distributed learning and optimization methods have been the focus of much research in recent years, with just a few examples including [25, 4, 2, 27, 1, 5, 13, 23, 16, 17, 8, 7, 9, 11, 20, 19, 3, 26]. Most of this work studied algorithms for this problem, which provide upper bounds on the required time and communication complexity.

In this paper, we take the opposite direction, and study what are the fundamental performance limitations in solving Eq. (1), under several different sets of assumptions. We identify cases where existing algorithms are already optimal (at least in the worst-case), as well as cases where room for further improvement is still possible.

Since a major constraint in distributed learning is communication, we focus on studying the amount of communication required to optimize Eq. (1) up to some desired accuracy $\epsilon$. More precisely, we consider the number of *communication rounds* that are required, where in each communication round the machines can generally broadcast to each other information linear in the problem's dimension $d$ (e.g. a point in $\mathcal{W}$ or a gradient). This applies to virtually all algorithms for large-scale learning we are aware of, where sending vectors and gradients is feasible, but computing and sending larger objects, such as Hessians ($d \times d$ matrices) is not.

Our results pertain to several possible settings (see Sec. 2 for precise definitions). First, we distinguish between the local functions being merely convex or strongly-convex, and whether they are smooth or not. These distinctions are standard in studying optimization algorithms for learning, and capture important properties such as the regularization and the type of loss function used. Second, we distinguish between a setting where the local functions are related – e.g., because they reflect statistical similarities in the data residing at different machines – and a setting where no relationship is assumed. For example, in the extreme case where data was split uniformly at random between machines, one can show that quantities such as the values, gradients and Hessians of the local functions differ only by $\delta = \mathcal{O}(1/\sqrt{n})$, where $n$ is the sample size per machine, due to concentration of measure effects. Such similarities can be used to speed up the optimization/learning process, as was done in e.g. [20, 26]. Both the $\delta$-related and the unrelated setting can be considered in a unified way, by letting $\delta$ be a parameter and studying the attainable lower bounds as a function of $\delta$. Our results can be summarized as follows:

- First, we define a mild structural assumption on the algorithm (which is satisfied by reasonable approaches we are aware of), which allows us to provide the lower bounds described below on the number of communication rounds required to reach a given suboptimality $\epsilon$.

  - When the local functions can be unrelated, we prove a lower bound of $\Omega(\sqrt{1/\lambda} \log(1/\epsilon))$ for smooth and $\lambda$-strongly convex functions, and $\Omega(\sqrt{1/\epsilon})$ for smooth convex functions. These lower bounds are matched by a straightforward distributed implementation of accelerated gradient descent. In particular, the results imply that many communication rounds may be required to get a high-accuracy solution, and moreover, that no algorithm satisfying our structural assumption would be better, even if we endow the local machines with unbounded computational power. For non-smooth functions, we show a lower bound of $\Omega(\sqrt{1/\lambda\epsilon})$ for $\lambda$-strongly convex functions, and $\Omega(1/\epsilon)$ for general convex functions. Although we leave a full derivation to future work, it seems these lower bounds can be matched in our framework by an algorithm combining acceleration and Moreau proximal smoothing of the local functions.

  - When the local functions are related (as quantified by the parameter $\delta$), we prove a communication round lower bound of $\Omega(\sqrt{\delta/\lambda} \log(1/\epsilon))$ for smooth and $\lambda$-strongly convex functions. For quadratics, this bound is matched by (up to constants and logarithmic factors) by the recently-proposed DISCO algorithm [26]. However, getting an optimal algorithm for general strongly convex and smooth functions in the $\delta$-related setting, let alone for non-smooth or non-strongly convex functions, remains open.

- We also study the attainable performance without posing any structural assumptions on the algorithm, but in the more restricted case where only a single round of communication is allowed. We prove that in a broad regime, the performance of any distributed algorithm may be no better than a 'trivial' algorithm which returns the minimizer of one of the local functions, as long as the number of bits communicated is less than $\Omega(d^2)$. Therefore, in our setting, no communication-efficient 1-round distributed algorithm can provide non-trivial performance in the worst case.

**Related Work**

There have been several previous works which considered lower bounds in the context of distributed learning and optimization, but to the best of our knowledge, none of them provide a similar type of results. Perhaps the most closely-related paper is [22], which studied the communication complexity of distributed optimization, and showed that $\Omega(d \log(1/\epsilon))$ bits of communication are necessary between the machines, for $d$-dimensional convex problems. However, in our setting this does not lead to any non-trivial lower bound on the number of communication rounds (indeed, just specifying a $d$-dimensional vector up to accuracy $\epsilon$ required $\mathcal{O}(d \log(1/\epsilon))$ bits). More recently, [2] considered lower bounds for certain types of distributed learning problems, but not convex ones in an agnostic

distribution-free framework. In the context of lower bounds for one-round algorithms, the results of [6] imply that $\Omega(d^2)$ bits of communication are required to solve linear regression in one round of communication. However, that paper assumes a different model than ours, where the function to be optimized is not split among the machines as in Eq. (1), where each $F_i$ is convex. Moreover, issues such as strong convexity and smoothness are not considered. [20] proves an impossibility result for a one-round distributed learning scheme, even when the local functions are not merely related, but actually result from splitting data uniformly at random between machines. On the flip side, that result is for a particular algorithm, and doesn't apply to any possible method.

Finally, we emphasize that distributed learning and optimization can be studied under many settings, including ones different than those studied here. For example, one can consider distributed learning on a stream of i.i.d. data [19, 7, 10, 8], or settings where the computing architecture is different, e.g. where the machines have a shared memory, or the function to be optimized is not split as in Eq. (1). Studying lower bounds in such settings is an interesting topic for future work.

## 2 Notation and Framework

The only vector and matrix norms used in this paper are the Euclidean norm and the spectral norm, respectively. $\mathbf{e}_j$ denotes the $j$-th standard unit vector. We let $\nabla G(\mathbf{w})$ and $\nabla^2 G(\mathbf{w})$ denote the gradient and Hessians of a function $G$ at $\mathbf{w}$, if they exist. $G$ is smooth (with parameter $L$) if it is differentiable and the gradient is $L$-Lipschitz. In particular, if $\mathbf{w}^* = \arg\min_{\mathbf{w} \in \mathcal{W}} G(\mathbf{w})$, then $G(\mathbf{w}) - G(\mathbf{w}^*) \leq \frac{L}{2} \|\mathbf{w} - \mathbf{w}^*\|^2$. $G$ is strongly convex (with parameter $\lambda$) if for any $\mathbf{w}, \mathbf{w}' \in \mathcal{W}$, $G(\mathbf{w}') \geq G(\mathbf{w}) + \langle \mathbf{g}, \mathbf{w}' - \mathbf{w} \rangle + \frac{\lambda}{2} \|\mathbf{w}' - \mathbf{w}\|^2$ where $\mathbf{g} \in \partial G(\mathbf{w}')$ is a subgradient of $G$ at $\mathbf{w}$. In particular, if $\mathbf{w}^* = \arg\min_{\mathbf{w} \in \mathcal{W}} G(\mathbf{w})$, then $G(\mathbf{w}) - G(\mathbf{w}^*) \geq \frac{\lambda}{2} \|\mathbf{w} - \mathbf{w}^*\|^2$. Any convex function is also strongly-convex with $\lambda = 0$. A special case of smooth convex functions are quadratics, where $G(\mathbf{w}) = \mathbf{w}^\top A \mathbf{w} + \mathbf{b}^\top \mathbf{w} + c$ for some positive semidefinite matrix $A$, vector $\mathbf{b}$ and scalar $c$. In this case, $\lambda$ and $L$ correspond to the smallest and largest eigenvalues of $A$.

We model the distributed learning algorithm as an iterative process, where in each round the machines may perform some local computations, followed by a communication round where each machine broadcasts a message to all other machines. We make no assumptions on the computational complexity of the local computations. After all communication rounds are completed, a designated machine provides the algorithm's output (possibly after additional local computation).

Clearly, without any assumptions on the number of bits communicated, the problem can be trivially solved in one round of communication (e.g. each machine communicates the function $F_i$ to the designated machine, which then solves Eq. (1). However, in practical large-scale scenarios, this is non-feasible, and the size of each message (measured by the number of bits) is typically on the order of $\tilde{\mathcal{O}}(d)$, enough to send a $d$-dimensional real-valued vector[1], such as points in the optimization domain or gradients, but not larger objects such as $d \times d$ Hessians.

In this model, our main question is the following: How many rounds of communication are *necessary* in order to solve problems such as Eq. (1) to some given accuracy $\epsilon$?

As discussed in the introduction, we first need to distinguish between different assumptions on the possible relation between the local functions. One natural situation is when no significant relationship can be assumed, for instance when the data is arbitrarily split or is gathered by each machine from statistically dissimilar sources. We denote this as the *unrelated* setting. However, this assumption is often unnecessarily pessimistic. Often the data allocation process is more random, or we can assume that the different data sources for each machine have statistical similarities (to give a simple example, consider learning from users' activity across a geographically distributed computing grid, each servicing its own local population). We will capture such similarities, in the context of quadratic functions, using the following definition:

**Definition 1.** *We say that a set of quadratic functions*

$$F_i(\mathbf{w}) := \mathbf{w}^\top A_i \mathbf{w} + \mathbf{b}_i \mathbf{w} + c_i, \qquad A_i \in \mathbb{R}^{d \times d}, \ \mathbf{b}_i \in \mathbb{R}^d, \ c_i \in \mathbb{R}$$

*are $\delta$-related, if for any $i, j \in \{1 \ldots k\}$, it holds that*

$$\|A_i - A_j\| \leq \delta, \ \|\mathbf{b}_i - \mathbf{b}_j\| \leq \delta, \ |c_i - c_j| \leq \delta$$

For example, in the context of linear regression with the squared loss over a bounded subset of $\mathbb{R}^d$, and assuming $mn$ data points with bounded norm are randomly and equally split among $m$ machines, it can be shown that the conditions above hold with $\delta = \mathcal{O}(1/\sqrt{n})$ [20]. The choice of $\delta$ provides us with a spectrum of learning problems ranked by difficulty: When $\delta = \Omega(1)$, this generally corresponds to the unrelated setting discussed earlier. When $\delta = \mathcal{O}(1/\sqrt{n})$, we get the situation typical of randomly partitioned data. When $\delta = 0$, then all the local functions have essentially the same minimizers, in which case Eq. (1) can be trivially solved with zero communication, just by letting one machine optimize its own local function. We note that although Definition 1 can be generalized to non-quadratic functions, we do not need it for the results presented here.

We end this section with an important remark. In this paper, we prove lower bounds for the $\delta$-related setting, which includes as a special case the commonly-studied setting of randomly partitioned data (in which case $\delta = \mathcal{O}(1/\sqrt{n})$). However, our bounds *do not apply* for random partitioning, since they use $\delta$-related constructions which do not correspond to randomly partitioned data. In fact, very recent work [12] has cleverly shown that for randomly partitioned data, and for certain reasonable regimes of strong convexity and smoothness, it is actually possible to get better performance than what is indicated by our lower bounds. However, this encouraging result crucially relies on the random partition property, and in parameter regimes which limit how much each data point needs to be "touched", hence preserving key statistical independence properties. We suspect that it may be difficult to improve on our lower bounds under substantially weaker assumptions.

## 3  Lower Bounds Using a Structural Assumption

In this section, we present lower bounds on the number of communication rounds, where we impose a certain mild structural assumption on the operations performed by the algorithm. Roughly speaking, our lower bounds pertain to a very large class of algorithms, which are based on linear operations involving points, gradients, and vector products with local Hessians and their inverses, as well as solving local optimization problems involving such quantities. At each communication round, the machines can share any of the vectors they have computed so far. Formally, we consider algorithms which satisfy the assumption stated below. For convenience, we state it for smooth functions (which are differentiable) and discuss the case of non-smooth functions in Sec. 3.2.

**Assumption 1.** *For each machine $j$, define a set $W_j \subset \mathbb{R}^d$, initially $W_j = \{\mathbf{0}\}$. Between communication rounds, each machine $j$ iteratively computes and adds to $W_j$ some finite number of points $\mathbf{w}$, each satisfying*

$$\gamma \mathbf{w} + \nu \nabla F_j(\mathbf{w}) \in \text{span} \left\{ \mathbf{w}' , \ \nabla F_j(\mathbf{w}') , \ (\nabla^2 F_j(\mathbf{w}') + D)\mathbf{w}'' , \ (\nabla^2 F_j(\mathbf{w}') + D)^{-1}\mathbf{w}'' \ \right| $$

$$\mathbf{w}', \mathbf{w}'' \in W_j \ , \ D \text{ diagonal} \ , \ \nabla^2 F_j(\mathbf{w}') \text{ exists} \ , \ (\nabla^2 F_j(\mathbf{w}') + D)^{-1} \text{ exists} \Big\}. \qquad (2)$$

*for some $\gamma, \nu \geq 0$ such that $\gamma + \nu > 0$. After every communication round, let $W_j := \cup_{i=1}^m W_i$ for all $j$. The algorithm's final output (provided by the designated machine $j$) is a point in the span of $W_j$.*

This assumption requires several remarks:

- Note that $W_j$ is not an explicit part of the algorithm: It simply includes all points computed by machine $j$ so far, or communicated to it by other machines, and is used to define the set of new points which the machine is allowed to compute.

- The assumption bears some resemblance – but is far weaker – than standard assumptions used to provide lower bounds for iterative optimization algorithms. For example, a common assumption (see [14]) is that each computed point $\mathbf{w}$ must lie in the span of the previous gradients. This corresponds to a special case of Assumption 1, where $\gamma = 1, \nu = 0$, and the span is only over gradients of previously computed points. Moreover, it also allows (for instance) exact optimization of each local function, which is a subroutine in some distributed algorithms (e.g. [27, 25]), by setting $\gamma = 0, \nu = 1$ and computing a point $\mathbf{w}$ satisfying $\gamma \mathbf{w} + \nu \nabla F_j(\mathbf{w}) = \mathbf{0}$. By allowing the span to include previous gradients, we also incorporate algorithms which perform optimization of the

local function plus terms involving previous gradients and points, such as [20], as well as algorithms which rely on local Hessian information and preconditioning, such as [26]. In summary, the assumption is satisfied by most techniques for black-box convex optimization that we are aware of. Finally, we emphasize that we do not restrict the number or computational complexity of the operations performed between communication rounds.

- The requirement that $\gamma, \nu \geq 0$ is to exclude algorithms which solve non-convex local optimization problems of the form $\min_{\mathbf{w}} F_j(\mathbf{w}) + \gamma \|\mathbf{w}\|^2$ with $\gamma < 0$, which are unreasonable in practice and can sometimes break our lower bounds.

- The assumption that $W_j$ is initially $\{\mathbf{0}\}$ (namely, that the algorithm starts from the origin) is purely for convenience, and our results can be easily adapted to any other starting point by shifting all functions accordingly.

The techniques we employ in this section are inspired by lower bounds on the iteration complexity of first-order methods for standard (non-distributed) optimization (see for example [14]). These are based on the construction of 'hard' functions, where each gradient (or subgradient) computation can only provide a small improvement in the objective value. In our setting, the dynamics are roughly similar, but the necessity of many gradient computations is replaced by many communication rounds. This is achieved by constructing suitable local functions, where at any time point no individual machine can 'progress' on its own, without information from other machines.

### 3.1 Smooth Local Functions

We begin by presenting a lower bound when the local functions $F_i$ are strongly-convex and smooth:

**Theorem 1.** *For any even number $m$ of machines, any distributed algorithm which satisfies Assumption 1, and for any $\lambda \in [0,1), \delta \in (0,1)$, there exist $m$ local quadratic functions over $\mathbb{R}^d$ (where $d$ is sufficiently large) which are $1$-smooth, $\lambda$-strongly convex, and $\delta$-related, such that if $\mathbf{w}^* = \arg\min_{\mathbf{w} \in \mathbb{R}^d} F(\mathbf{w})$, then the number of communication rounds required to obtain $\hat{\mathbf{w}}$ satisfying $F(\hat{\mathbf{w}}) - F(\mathbf{w}^*) \leq \epsilon$ (for any $\epsilon > 0$) is at least*

$$\frac{1}{4}\left(\sqrt{1 + \delta\left(\frac{1}{\lambda} - 1\right)} - 1\right)\log\left(\frac{\lambda\|\mathbf{w}^*\|^2}{4\epsilon}\right) - \frac{1}{2} = \Omega\left(\sqrt{\frac{\delta}{\lambda}}\log\left(\frac{\lambda\|\mathbf{w}^*\|^2}{\epsilon}\right)\right)$$

*if $\lambda > 0$, and at least $\sqrt{\frac{3\delta}{32\epsilon}}\|\mathbf{w}^*\| - 2$ if $\lambda = 0$.*

The assumption of $m$ being even is purely for technical convenience, and can be discarded at the cost of making the proof slightly more complex. Also, note that $m$ does not appear explicitly in the bound, but may appear implicitly, via $\delta$ (for example, in a statistical setting $\delta$ may depend on the number of data points per machine, and may be larger if the same dataset is divided to more machines).

Let us contrast our lower bound with some existing algorithms and guarantees in the literature. First, regardless of whether the local functions are similar or not, we can always simulate any gradient-based method designed for a single machine, by iteratively computing gradients of the local functions, and performing a communication round to compute their average. Clearly, this will be a gradient of the objective function $F(\cdot) = \frac{1}{m}\sum_{i=1}^{m} F_i(\cdot)$, which can be fed into any gradient-based method such as gradient descent or accelerated gradient descent [14]. The resulting number of required communication rounds is then equal to the number of iterations. In particular, using accelerated gradient descent for smooth and $\lambda$-strongly convex functions yields a round complexity of $\mathcal{O}(\sqrt{1/\lambda}\log(\|\mathbf{w}^*\|^2/\epsilon))$, and $\mathcal{O}(\|\mathbf{w}^*\|\sqrt{1/\epsilon})$ for smooth convex functions. This matches our lower bound (up to constants and log factors) when the local functions are unrelated ($\delta = \Omega(1)$).

When the functions are related, however, the upper bounds above are highly sub-optimal: Even if the local functions are completely identical, and $\delta = 0$, the number of communication rounds will remain the same as when $\delta = \Omega(1)$. To utilize function similarity while guaranteeing arbitrary small $\epsilon$, the two most relevant algorithms are DANE [20], and the more recent DISCO [26]. For smooth and $\lambda$-strongly convex functions, which are either quadratic or satisfy a certain self-concordance condition, DISCO achieves $\tilde{\mathcal{O}}(1+\sqrt{\delta/\lambda})$ round complexity ([26, Thm.2]), which matches our lower bound in terms of dependence on $\delta, \lambda$. However, for non-quadratic losses, the round complexity

bounds are somewhat worse, and there are no guarantees for strongly convex and smooth functions which are not self-concordant. Thus, the question of the optimal round complexity for such functions remains open.

The full proof of Thm. 1 appears in the supplementary material, and is based on the following idea: For simplicity, suppose we have two machines, with local functions $F_1, F_2$ defined as follows,

$$F_1(\mathbf{w}) = \frac{\delta(1-\lambda)}{4}\mathbf{w}^\top A_1 \mathbf{w} - \frac{\delta(1-\lambda)}{2}\mathbf{e}_1^\top \mathbf{w} + \frac{\lambda}{2}\|\mathbf{w}\|^2 \tag{3}$$

$$F_2(\mathbf{w}) = \frac{\delta(1-\lambda)}{4}\mathbf{w}^\top A_2 \mathbf{w} + \frac{\lambda}{2}\|\mathbf{w}\|^2, \quad \text{where}$$

$$A_1 = \begin{bmatrix} 1 & 0 & 0 & 0 & 0 & 0 & \dots \\ 0 & 1 & -1 & 0 & 0 & 0 & \dots \\ 0 & -1 & 1 & 0 & 0 & 0 & \dots \\ 0 & 0 & 0 & 1 & -1 & 0 & \dots \\ 0 & 0 & 0 & -1 & 1 & 0 & \dots \\ \vdots & \vdots & \vdots & \vdots & \vdots & \vdots & \vdots \end{bmatrix}, \quad A_2 = \begin{bmatrix} 1 & -1 & 0 & 0 & 0 & 0 & \dots \\ -1 & 1 & 0 & 0 & 0 & 0 & \dots \\ 0 & 0 & 1 & -1 & 0 & 0 & \dots \\ 0 & 0 & -1 & 1 & 0 & 0 & \dots \\ 0 & 0 & 0 & 0 & 1 & -1 & \dots \\ 0 & 0 & 0 & 0 & -1 & 1 & \dots \\ \vdots & \vdots & \vdots & \vdots & \vdots & \vdots & \vdots \end{bmatrix}$$

It is easy to verify that for $\delta, \lambda \leq 1$, both $F_1(\mathbf{w})$ and $F_2(\mathbf{w})$ are 1-smooth and $\lambda$-strongly convex, as well as $\delta$-related. Moreover, the optimum of their average is a point $\mathbf{w}^*$ with non-zero entries at all coordinates. However, since each local functions has a block-diagonal quadratic term, it can be shown that for any algorithm satisfying Assumption 1, after $T$ communication rounds, the points computed by the two machines can only have the first $T + 1$ coordinates non-zero. No machine will be able to further 'progress' on its own, and cause additional coordinates to become non-zero, without another communication round. This leads to a lower bound on the optimization error which depends on $T$, resulting in the theorem statement after a few computations.

## 3.2 Non-smooth Local Functions

Remaining in the framework of algorithms satisfying Assumption 1, we now turn to discuss the situation where the local functions are not necessarily smooth or differentiable. For simplicity, our formal results here will be in the unrelated setting, and we only informally discuss their extension to a $\delta$-related setting (in a sense relevant to non-smooth functions). Formally defining $\delta$-related non-smooth functions is possible but not altogether trivial, and is therefore left to future work.

We adapt Assumption 1 to the non-smooth case, by allowing gradients to be replaced by arbitrary subgradients at the same points. Namely, we replace Eq. (2) by the requirement that for some $\mathbf{g} \in \partial F_j(\mathbf{w})$, and $\gamma, \nu \geq 0, \gamma + \nu > 0$,

$$\gamma\mathbf{w} + \nu\mathbf{g} \in \text{span}\left\{ \mathbf{w}' , \mathbf{g}' , (\nabla^2 F_j(\mathbf{w}') + D)\mathbf{w}'' , (\nabla^2 F_j(\mathbf{w}') + D)^{-1}\mathbf{w}'' \right| $$

$$\left. \mathbf{w}', \mathbf{w}'' \in W_j , \mathbf{g}' \in \partial F_j(\mathbf{w}') , D \text{ diagonal} , \nabla^2 F_j(\mathbf{w}') \text{ exists} , (\nabla^2 F_j(\mathbf{w}') + D)^{-1} \text{ exists} \right\}.$$

The lower bound for this setting is stated in the following theorem.

**Theorem 2.** *For any even number $m$ of machines, any distributed optimization algorithm which satisfies Assumption 1, and for any $\lambda \geq 0$, there exist $\lambda$-strongly convex $(1+\lambda)$-Lipschitz continuous convex local functions $F_1(\mathbf{w})$ and $F_2(\mathbf{w})$ over the unit Euclidean ball in $\mathbb{R}^d$ (where $d$ is sufficiently large), such that if $\mathbf{w}^* = \arg\min_{\mathbf{w}:\|\mathbf{w}\|\leq 1} F(\mathbf{w})$, the number of communication rounds required to obtain $\hat{\mathbf{w}}$ satisfying $F(\hat{\mathbf{w}}) - F(\mathbf{w}^*) \leq \epsilon$ (for any sufficiently small $\epsilon > 0$) is $\frac{1}{8\epsilon} - 2$ for $\lambda = 0$, and $\sqrt{\frac{1}{16\lambda\epsilon}} - 2$ for $\lambda > 0$.*

As in Thm. 1, we note that the assumption of even $m$ is for technical convenience.

This theorem, together with Thm. 1, implies that both strong convexity and smoothness are necessary for the number of communication rounds to scale logarithmically with the required accuracy $\epsilon$. We emphasize that this is true even if we allow the machines unbounded computational power, to perform arbitrarily many operations satisfying Assumption 1. Moreover, a preliminary analysis

indicates that performing accelerated gradient descent on smoothed versions of the local functions (using Moreau proximal smoothing, e.g. [15, 24]), can match these lower bounds up to log factors[2]. We leave a full formal derivation (which has some subtleties) to future work.

The full proof of Thm. 2 appears in the supplementary material. The proof idea relies on the following construction: Assume that we fix the number of communication rounds to be $T$, and (for simplicity) that $T$ is even and the number of machines is 2. Then we use local functions of the form

$$F_1(\mathbf{w}) = \frac{1}{\sqrt{2}}\,|b - w_1| + \frac{1}{\sqrt{2(T+2)}}\left(|w_2 - w_3| + |w_4 - w_5| + \cdots + |w_T - w_{T+1}|\right) + \frac{\lambda}{2}\,\|\mathbf{w}\|^2$$

$$F_2(\mathbf{w}) = \frac{1}{\sqrt{2(T+2)}}\left(|w_1 - w_2| + |w_3 - w_4| + \cdots + |w_{T+1} - w_{T+2}|\right) + \frac{\lambda}{2}\,\|\mathbf{w}\|^2,$$

where $b$ is a suitably chosen parameter. It is easy to verify that both local functions are $\lambda$-strongly convex and $(1 + \lambda)$-Lipschitz continuous over the unit Euclidean ball. Similar to the smooth case, we argue that after $T$ communication rounds, the resulting points $\mathbf{w}$ computed by machine 1 will be non-zero only on the first $T + 1$ coordinates, and the points $\mathbf{w}$ computed by machine 2 will be non-zero only on the first $T$ coordinates. As in the smooth case, these functions allow us to 'control' the progress of any algorithm which satisfies Assumption 1.

Finally, although the result is in the unrelated setting, it is straightforward to have a similar construction in a '$\delta$-related' setting, by multiplying $F_1$ and $F_2$ by $\delta$. The resulting two functions have their gradients and subgradients at most $\delta$-different from each other, and the construction above leads to a lower bound of $\Omega(\delta/\epsilon)$ for convex Lipschitz functions, and $\Omega(\delta\sqrt{1/\lambda\epsilon})$ for $\lambda$-strongly convex Lipschitz functions. In terms of upper bounds, we are actually unaware of any relevant algorithm in the literature adapted to such a setting, and the question of attainable performance here remains wide open.

## 4 One Round of Communication

In this section, we study what lower bounds are attainable without any kind of structural assumption (such as Assumption 1). This is a more challenging setting, and the result we present will be limited to algorithms using a single round of communication round. We note that this still captures a realistic non-interactive distributed computing scenario, where we want each machine to broadcast a single message, and a designated machine is then required to produce an output. In the context of distributed optimization, a natural example is a one-shot averaging algorithm, where each machine optimizes its own local data, and the resulting points are averaged (e.g. [27, 25]).

Intuitively, with only a single round of communication, getting an arbitrarily small error $\epsilon$ may be infeasible. The following theorem establishes a lower bound on the attainable error, depending on the strong convexity parameter $\lambda$ and the similarity measure $\delta$ between the local functions, and compares this with a 'trivial' zero-communication algorithm, which just returns the optimum of a single local function:

**Theorem 3.** *For any even number $m$ of machines, any dimension $d$ larger than some numerical constant, any $\delta \geq 3\lambda > 0$, and any (possibly randomized) algorithm which communicates at most $d^2/128$ bits in a single round of communication, there exist $m$ quadratic functions over $\mathbb{R}^d$, which are $\delta$-related, $\lambda$-strongly convex and $9\lambda$-smooth, for which the following hold for some positive numerical constants $c, c'$:*

- *The point $\hat{\mathbf{w}}$ returned by the algorithm satisfies*

$$\mathbb{E}\left[F(\hat{\mathbf{w}}) - \min_{\mathbf{w}\in\mathbb{R}^d} F(\mathbf{w})\right] \geq c\frac{\delta^2}{\lambda}$$

*in expectation over the algorithm's randomness.*

- *For any machine $j$, if $\hat{\mathbf{w}}_j = \arg\min_{\mathbf{w}\in\mathbb{R}^d} F_j(\mathbf{w})$, then $F(\hat{\mathbf{w}}_j) - \min_{\mathbf{w}\in\mathbb{R}^d} F(\mathbf{w}) \leq c'\delta^2/\lambda$.*

The theorem shows that unless the communication budget is extremely large (quadratic in the dimension), there are functions which cannot be optimized to non-trivial accuracy in one round of communication, in the sense that the same accuracy (up to a universal constant) can be obtained with a 'trivial' solution where we just return the optimum of a single local function. This complements an earlier result in [20], which showed that a *particular* one-round algorithm is no better than returning the optimum of a local function, under the stronger assumption that the local functions are not merely $\delta$-related, but are actually the average loss over some randomly partitioned data.

The full proof appears in the supplementary material, but we sketch the main ideas below. As before, focusing on the case of two machines, and assuming machine 2 is responsible for providing the output, we use

$$F_1(\mathbf{w}) = 3\lambda\mathbf{w}^\top \left( \left( I + \frac{1}{2c\sqrt{d}}M \right)^{-1} - \frac{1}{2}I \right)\mathbf{w}$$

$$F_2(\mathbf{w}) = \frac{3\lambda}{2}\|\mathbf{w}\|^2 - \delta\mathbf{e}_j,$$

where $M$ is essentially a randomly chosen $\{-1,+1\}$-valued $d \times d$ symmetric matrix with spectral norm at most $c\sqrt{d}$, and $c$ is a suitable constant. These functions can be shown to be $\delta$-related as well as $\lambda$-strongly convex. Moreover, the optimum of $F(\mathbf{w}) = \frac{1}{2}(F_1(\mathbf{w}) + F_2(\mathbf{w}))$ equals

$$\mathbf{w}^* = \frac{\delta}{6\lambda}\left( I + \frac{1}{2c\sqrt{d}}M \right)\mathbf{e}_j.$$

Thus, we see that the optimal point $\mathbf{w}^*$ depends on the $j$-th column of $M$. Intuitively, the machines need to approximate this column, and this is the source of hardness in this setting: Machine 1 knows $M$ but not $j$, yet needs to communicate to machine 2 enough information to construct its $j$-th column. However, given a communication budget much smaller than the size of $M$ (which is $d^2$), it is difficult to convey enough information on the $j$-th column without knowing what $j$ is. Carefully formalizing this intuition, and using some information-theoretic tools, allows us to prove the first part of Thm. 3. Proving the second part of Thm. 3 is straightforward, using a few computations.

## 5   Summary and Open Questions

In this paper, we studied lower bounds on the number of communication rounds needed to solve distributed convex learning and optimization problems, under several different settings. Our results indicate that when the local functions are unrelated, then regardless of the local machines' computational power, many communication rounds may be necessary (scaling polynomially with $1/\epsilon$ or $1/\lambda$), and that the worst-case optimal algorithm (at least for smooth functions) is just a straightforward distributed implementation of accelerated gradient descent. When the functions are related, we show that the optimal performance is achieved by the algorithm of [26] for quadratic and strongly convex functions, but designing optimal algorithms for more general functions remains open. Beside these results, which required a certain mild structural assumption on the algorithm employed, we also provided an assumption-free lower bound for one-round algorithms, which implies that even for strongly convex quadratic functions, such algorithms can sometimes only provide trivial performance.

Besides the question of designing optimal algorithms for the remaining settings, several additional questions remain open. First, it would be interesting to get assumption-free lower bounds for algorithms with multiple rounds of communication. Second, our work focused on *communication* complexity, but in practice the *computational* complexity of the local computations is no less important. Thus, it would be interesting to understand what is the attainable performance with simple, runtime-efficient algorithms. Finally, it would be interesting to study lower bounds for other distributed learning and optimization scenarios.

**Acknowledgments:**   This research is supported in part by an FP7 Marie Curie CIG grant, the Intel ICRI-CI Institute, and Israel Science Foundation grant 425/13. We thank Nati Srebro for several helpful discussions and insights.

## Footnotes

[1]The $\tilde{\mathcal{O}}$ hides constants and factors logarithmic in the required accuracy of the solution. The idea is that we can represent real numbers up to some arbitrarily high machine precision, enough so that finite-precision issues are not a problem.

[2]Roughly speaking, for any $\gamma > 0$, this smoothing creates a $\frac{1}{\gamma}$-smooth function which is $\gamma$-close to the original function. Plugging these into the guarantees of accelerated gradient descent and tuning $\gamma$ yields our lower bounds. Note that, in order to execute this algorithm each machine must be sufficiently powerful to obtain the gradient of the Moreau envelope of its local function, which is indeed the case in our framework.

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
