[Supplementary Material]

# A   Proofs

## A.1   Proof of Thm. 1

The proof of the theorem is based on splitting the machines into two sub-groups of the same size, each of which is assigned with a finite dimensional restriction of $F_1$ and $F_2$ (see Eq. (3)), and tracing the maximal number of non-zero coordinates for vectors in $W_j$, the set of feasible points.

Recall that $F_i$ are defined as follows:

$$F_1(\mathbf{w}) = \frac{\delta(1-\lambda)}{4} \mathbf{w}^\top A_1 \mathbf{w} - \frac{\delta(1-\lambda)}{2} \mathbf{e}_1^\top \mathbf{w} + \frac{\lambda}{2} \|\mathbf{w}\|^2$$

$$F_2(\mathbf{w}) = \frac{\delta(1-\lambda)}{4} \mathbf{w}^\top A_2 \mathbf{w} + \frac{\lambda}{2} \|\mathbf{w}\|^2, \quad \text{where}$$

$$A_1 = \begin{bmatrix} 1 & 0 & 0 & 0 & 0 & 0 & \dots \\ 0 & 1 & -1 & 0 & 0 & 0 & \dots \\ 0 & -1 & 1 & 0 & 0 & 0 & \dots \\ 0 & 0 & 0 & 1 & -1 & 0 & \dots \\ 0 & 0 & 0 & -1 & 1 & 0 & \dots \\ \vdots & \vdots & \vdots & \vdots & \vdots & \vdots & \vdots \end{bmatrix}, \quad A_2 = \begin{bmatrix} 1 & -1 & 0 & 0 & 0 & 0 & \dots \\ -1 & 1 & 0 & 0 & 0 & 0 & \dots \\ 0 & 0 & 1 & -1 & 0 & 0 & \dots \\ 0 & 0 & -1 & 1 & 0 & 0 & \dots \\ 0 & 0 & 0 & 0 & 1 & -1 & \dots \\ 0 & 0 & 0 & 0 & -1 & 1 & \dots \\ \vdots & \vdots & \vdots & \vdots & \vdots & \vdots & \vdots \end{bmatrix}$$

Formally speaking, we consider the matrices $A_1, A_2$ as infinite in size, so that each $F_i$ is defined over $\ell^2(\mathbb{R})$, the space of square-summable sequences. To derive lower bounds in $\mathbb{R}^d$, we consider the following restrictions of $F_i$ and $F$:

$$[F_i]_d(\mathbf{w}) := F_i(w_1, w_2, \dots, w_d, 0, 0, \dots), \quad \mathbf{w} \in \mathbb{R}^d$$

$$[F]_d(\mathbf{w}) := \frac{[F_1]_d(\mathbf{w}) + [F_2]_d(\mathbf{w})}{2}$$

Note that $[F_i]_d(\mathbf{w})$ and $[F]_d(\mathbf{w})$ produce the same values as $F_i(\mathbf{w})$ and $F(\mathbf{w})$ do for vectors such that $\mathbf{w}_i = 0$ for all $i \geq d$. Similarly, we define the $d \times d$ leading principal submatrix of $A_i$ by $[A_i]_d$.

We assign half of the machines with $[F_1]_d$, and the other half with $[F_2]_d$. To prove the theorem, we need the following lemma, which formalizes the intuition described in the main paper. Let

$$E_{0,d} = \{\mathbf{0}\} \quad, \quad E_{T,d} = \text{span}\{\mathbf{e}_{1,d}, \dots, \mathbf{e}_{T,d}\},$$

where $\mathbf{e}_{i,d} \in \mathbb{R}^d$ denote the standard unit vectors. Then, the following holds:

**Lemma 1.** *Suppose all the sets of feasible points satisfy $W_j \subseteq E_{T,d}$ for some $T \leq d-1$, then under assumption 1, right after the next communication round we have $W_j \subseteq E_{T+1,d}$.*

*Proof.* Recall that by Assumption 1, each machine can compute new points $\mathbf{w}$ that satisfy the following for some $\gamma, \nu \geq 0$ such that $\gamma + \nu > 0$:

$$\gamma \mathbf{w} + \nu \nabla[F_i]_d(\mathbf{w}) \in \text{span} \Big\{ \mathbf{w}', \ \nabla[F_i]_d(\mathbf{w}'), \ (\nabla^2[F_i]_d(\mathbf{w}') + D)\mathbf{w}'', \ (\nabla^2[F_i]_d(\mathbf{w}') + D)^{-1}\mathbf{w}'' \ \Big|$$

$$\mathbf{w}', \mathbf{w}'' \in W_j \ , \ D \text{ diagonal} \ , \ \nabla^2[F_i]_d(\mathbf{w}') \text{ exists} \ , \ (\nabla^2[F_i]_d(\mathbf{w}') + D)^{-1} \text{ exists} \Big\}.$$

We now analyze the state of the sets of feasible points prior to the next communication round. Assume that $T$ is an odd number, i.e., assume $T = 2k+1$ for some $k = 0, 1, \dots$. The proof for the case where $T$ is even follows similar lines. Note that for any $\mathbf{w}', \mathbf{w}'' \in W_j$, we have

$$\nabla[F_1]_d(\mathbf{w}') = \frac{\delta(1-\lambda)}{2}[A_1]_d \mathbf{w}' - \frac{\delta(1-\lambda)}{2}\mathbf{e}_1 + \frac{\lambda}{2}\mathbf{w}' \subseteq E_{2k+1,d}$$

$$(\nabla^2[F_1]_d(\mathbf{w}') + D)\mathbf{w}'' = \left(\frac{\delta(1-\lambda)}{2}[A_1]_d + D + \lambda I\right)\mathbf{w}'' \subseteq E_{2k+1,d}$$

$$(\nabla^2[F_1]_d(\mathbf{w}') + D)^{-1}\mathbf{w}'' = \left(\frac{\delta(1-\lambda)}{2}[A_1]_d + D + \lambda I\right)^{-1}\mathbf{w}'' \subseteq E_{2k+1,d}$$

For any viable diagonal matrix D. Therefore, since $W_j \subseteq E_{2k+1,d}$, we have that the first point generated by machines which hold $[F_1]_d(\mathbf{w})$ must satisfy

$$\gamma \mathbf{w} + \nu \nabla [F_1]_d(\mathbf{w}) \in E_{2k+1,d}$$

for $\gamma, \nu$ as stated in the assumption. That is,

$$\left( \frac{\delta(1-\lambda)}{2} \nu [A_1]_d + (\gamma + \frac{\nu\lambda}{2})I \right) \mathbf{w} - \frac{\delta(1-\lambda)}{2} \mathbf{e}_1 \in E_{2k+1,d}$$

Which implies,

$$\underbrace{\left( \frac{\delta(1-\lambda)}{2} \nu [A_1]_d + (\gamma + \frac{\nu\lambda}{2})I \right)}_{H} \mathbf{w} \in E_{2k+1,d}$$

Since $[A_1]_d$ is positive semidefinite, it holds that $H$ is invertible. Also, $[A_1]_d, H$ and $H^{-1}$ admit the same partitions into $1 \times 1$ and $2 \times 2$ blocks on the diagonal, thus $H^{-1} E_{2k+1,d} \subseteq E_{2k+1,d}$, yielding $\mathbf{w} \in E_{2k+1,d}$. Inductively extending the latter argument shows that, in the absence of any communication rounds, all the machines whose local function is $[F_1]_d(\mathbf{w})$ are 'stuck' in $E_{2k+1,d}$.

As for machines which contain $[F_2]_d(\mathbf{w})$, we have that for all $\mathbf{w}', \mathbf{w}'' \in W_j$

$$\nabla [F_2]_d(\mathbf{w}') = \frac{\delta(1-\lambda)}{2} [A_2]_d \mathbf{w}' + \frac{\lambda}{2} \mathbf{w}' \subseteq E_{2k+2,d}$$

$$(\nabla^2 [F_2]_d(\mathbf{w}') + D)\mathbf{w}'' = \left( \frac{\delta(1-\lambda)}{2} [A_2]_d + D + \lambda I \right) \mathbf{w}'' \subseteq E_{2k+2,d}$$

$$(\nabla^2 [F_2]_d(\mathbf{w}') + D)^{-1}\mathbf{w}'' = \left( \frac{\delta(1-\lambda)}{2} [A_2]_d + D + \lambda I \right)^{-1} \mathbf{w}'' \subseteq E_{2k+2,d}$$

For any viable diagonal matrix D. Therefore, the first generated point by these machines must satisfy,

$$\gamma \mathbf{w} + \nu \nabla [F_1]_d(\mathbf{w}) \in E_{2k+2,d}$$

for appropriate $\gamma, \nu$. Hence,

$$\left( \frac{\delta(1-\lambda)}{2} \nu [A_2]_d + (\gamma + \frac{\nu\lambda}{2})I \right) \mathbf{w} \in E_{2k+2,d}$$

Similarly to the previous case this implies that $\mathbf{w} \in E_{2k+2,d}$. It is now left to show that these machines cannot make further progress beyond $E_{2k+2,d}$ without communicating. To see this, note that for all $\mathbf{w}', \mathbf{w}'' \in E_{2k+2,d}$ we have,

$$\nabla [F_2]_d(\mathbf{w}') = \frac{\delta(1-\lambda)}{2} [A_2]_d \mathbf{w}' + \frac{\lambda}{2} \mathbf{w}' \subseteq E_{2k+2,d}$$

$$(\nabla^2 [F_2]_d(\mathbf{w}') + D)\mathbf{w}'' = \left( \frac{\delta(1-\lambda)}{2} [A_2]_d + D + \lambda I \right) \mathbf{w}'' \subseteq E_{2k+2,d}$$

$$(\nabla^2 [F_2]_d(\mathbf{w}') + D)^{-1}\mathbf{w}'' = \left( \frac{\delta(1-\lambda)}{2} [A_2]_d + DI + \lambda I \right)^{-1} \mathbf{w}'' \subseteq E_{2k+2,d}$$

This means that all the points which are generated subsequently also lie in $E_{2k+2,d}$, i.e., without communicating , machines whose local function is $[F_2]_d(\mathbf{w})$ are stuck in $E_{2k+2,d}$. Finally, executing a communication round updates all the sets of feasible points to be $W_j := E_{2k+2,d}$. □

The following is a direct consequence of a recursive application of Lemma 1.

**Corollary 1.** *Under assumption 1, after $T \leq d - 1$ communication rounds we have*

$$W_j \subseteq E_{T+1}, \quad j \in \{1, \dots, m\}$$

With this corollary in hand, we now turn to prove the main result. First, we compute the minimizer of the average function $F(\mathbf{w}) = \frac{\frac{m}{2} F_1(\mathbf{w}) + \frac{m}{2} F_2(\mathbf{w})}{m}$ in $\ell^2(\mathbb{R})$, denoted by $\mathbf{w}^*$, whose form for even number of machines is simply:

$$F(\mathbf{w}) = \frac{\delta(1-\lambda)}{8} \mathbf{w}^\top (A_1 + A_2) \, \mathbf{w} - \frac{\delta(1-\lambda)}{4} \mathbf{e}_1^\top \mathbf{w} + \frac{\lambda}{2} \|\mathbf{w}\|^2$$

By first-order optimality condition for smooth convex functions, we have

$$\left( \frac{\delta(1-\lambda)}{4} (A_1 + A_2) + \lambda I \right) \mathbf{w}^* - \frac{\delta(1-\lambda)}{4} \mathbf{e}_1 = 0,$$

or equivalently,

$$\left( A_1 + A_2 + \frac{4\lambda}{\delta(1-\lambda)} I \right) \mathbf{w}^* = \mathbf{e}_1$$

whose coordinate form is as follows

$$\left( 2 + \frac{4\lambda}{\delta(1-\lambda)} \right) \mathbf{w}^*[1] - \mathbf{w}^*[2] = 1 \qquad (4)$$

$$\forall k, \quad \mathbf{w}^*[k+1] - \left( 2 + \frac{4\lambda}{\delta(1-\lambda)} \right) \mathbf{w}^*[k] + \mathbf{w}^*[k-1] = 0. \qquad (5)$$

The optimal solution can be now realized as a geometric sequence $(\zeta^k)_{k=1}^\infty$ for some $\zeta$ as follows: By Eq. (5), we must have

$$\zeta^2 - \left( 2 + \frac{4\lambda}{\delta(1-\lambda)} \right) \zeta + 1 = 0,$$

with the smallest root being

$$\zeta = \frac{2 + \frac{4\lambda}{\delta(1-\lambda)} - \sqrt{\left(2 + \frac{4\lambda}{\delta(1-\lambda)}\right)^2 - 4}}{2} = 1 + \frac{2\lambda}{\delta(1-\lambda)} - \sqrt{\left(1 + \frac{2\lambda}{\delta(1-\lambda)}\right)^2 - 1} \qquad (6)$$

Therefore, this choice of $\zeta$ satisfies Eq. (5), and it is straightforward to verify that it also satisfies Eq. (4), hence $\mathbf{w}^*$ indeed equals $(\zeta^k)_{k=1}^\infty$. It will be convenient to denote a continuous range of coordinates of $(\zeta^k)_{k=1}^\infty$ by $\boldsymbol{\zeta}_{a:b}$ where $a \in \mathbb{N}$ and $b \in \mathbb{N} \cup \infty$. Also, using the following inequality which holds for $x > 1$

$$x - \sqrt{x^2 - 1} \geq \exp\left( \frac{-2}{\sqrt{\frac{x+1}{x-1}} - 1} \right), \quad x > 1$$

together with Eq. (6) yields

$$\zeta \geq \exp\left( \frac{-2}{\sqrt{\delta(1/\lambda - 1) + 1} - 1} \right) \qquad (7)$$

We now use this computation (with respect to $F_1, F_2$) to find the minimizer of $[F]_d$, defined as the average function of the finite-dimensional restrictions $[F_1]_d, [F_2]_d$ actually handed to the machines. Fix $d \in \mathbb{N}$ and denote the corresponding minimizer by

$$\mathbf{w}_d^* = \arg \min_{\mathbf{w} \in \mathbb{R}^d} [F]_d(\mathbf{w})$$

Let $\mathbf{w}_T$ be some point which was obtained after $T \leq d - 2$ communication rounds. To bound the sub-optimality of $\mathbf{w}_T$ from below, observe that

$$[F]_d(\mathbf{w}_T) - [F]_d(\mathbf{w}_d^*) \geq [F]_d(\mathbf{w}_T) - [F]_d(\boldsymbol{\zeta}_{1:d-1})$$
$$= [F]_d(\mathbf{w}_T) - F(\mathbf{w}^*) + F(\mathbf{w}^*) - [F]_d(\boldsymbol{\zeta}_{1:d-1})$$
$$= \underbrace{F(\mathbf{w}_T) - F(\mathbf{w}^*)}_{A} + \underbrace{F(\mathbf{w}^*) - F(\boldsymbol{\zeta}_{1:d-1})}_{B}$$

where the last equality follows from Corollary (1), according to which all the coordinates of $\mathbf{w}_T$, except for the first $T + 1 \leq d - 1$, must vanish. To bound the $A$ term, note that

$$\|\mathbf{w}_T - \mathbf{w}^*\|^2 \geq \sum_{t=T+2}^{\infty} \zeta^{2t} = \zeta^{2(T+1)} \sum_{t=1}^{\infty} \zeta^{2t} = \zeta^{2(T+1)} \|\mathbf{w}^*\|^2$$

The fact that $F(\mathbf{w})$ is $\lambda$-strongly convex implies

$$F(\mathbf{w}_T) - F(\mathbf{w}^*) \geq \frac{\lambda}{2} \|\mathbf{w}_T - \mathbf{w}^*\|^2 \geq \frac{\lambda \zeta^{2(T+1)}}{2} \|\mathbf{w}^*\|^2 .$$

Inequality (7) yields

$$F(\mathbf{w}_T) - F(\mathbf{w}^*) \geq \frac{\lambda}{2} \exp\left( \frac{-4T - 4}{\sqrt{\delta(1/\lambda - 1) + 1} - 1} \right) \|\mathbf{w}^*\|^2$$

To bound the $B$ term from below, note that since $F$ is 1-smooth we have

$$F(\mathbf{w}^*) - F(\boldsymbol{\zeta}_{1:d-1}) \geq -\frac{1}{2} \|\mathbf{w}^* - \boldsymbol{\zeta}_{1:d-1}\|^2 = -\frac{\zeta^{2(d-1)}}{2} \sum_{t=1}^{\infty} \zeta^{2t} = -\frac{\zeta^{2(d-1)}}{2} \|\mathbf{w}^*\|^2$$

Combining both lower bounds for the terms $A$ and $B$, we get for any $T \leq d - 2$

$$[F]_d(\mathbf{w}_T) - [F]_d(\mathbf{w}_d^*) \geq \left( \frac{\lambda}{2} \exp\left( \frac{-4T - 4}{\sqrt{\delta(1/\lambda - 1) + 1} - 1} \right) - \frac{\zeta^{2(d-1)}}{2} \right) \|\mathbf{w}^*\|^2$$

Picking $d$ sufficiently large, and considering how large the number of communication rounds $T$ must be to make this lower bound less than $\epsilon$, we get

$$T \geq \frac{\sqrt{\delta(1/\lambda - 1) + 1} - 1}{4} \ln\left( \frac{\lambda \|\mathbf{w}^*\|^2}{4\epsilon} \right) - 1.$$

It is worth mentioning that by computing the exact minimizers of $[F_i]_d$ one may derive a lower bound such that the choice of $d$ does not depend on the parameters of the problem, except for the number of communication rounds. Nevertheless, such analysis requires a more involved reasoning which we find unnecessary for stating our results.

For the non-strongly convex case, where $\lambda = 0$, using Corollary 1 and a similar analysis (virtually identical to the proof of Theorem 2.1.7 in [14]), we have that if $T \leq \frac{1}{2}(d - 1)$, then

$$F(\mathbf{w}_T) - F(\mathbf{w}^*) \geq \frac{3\delta \|w^*\|^2}{32(T + 2)^2}$$

Therefore, to obtain an $\epsilon$-suboptimal solution for this case, we must have at least

$$\sqrt{\frac{3\delta}{32\epsilon}} \|\mathbf{w}^*\| - 2$$

communication rounds, for sufficiently small $\epsilon$.

## A.2   Proof of Thm. 2

We construct two types of local functions, and provide one of them to $m/2$ of the machines, and the other function to the other $m/2$ machines, in some arbitrary order. In this case, the average function is simply the average of the two types of local functions.

We will first prove the theorem statement in the strongly convex case, where $\lambda > 0$ is given, and then explain how to extract from it the result in the non strongly convex case.

Fix a natural number $k$ and some $b \in [0, 1/\sqrt{k}]$, to be specified later. We define the following local function over the unit ball:

$$F_{1,k}(\mathbf{w}) = \frac{1}{\sqrt{2}} |b - w[1]| + \frac{1}{\sqrt{2k}} (|w[2] - w[3]| + |w[4] - w[5]| + \cdots + |w[k - 2] - w[k - 1]|) + \frac{\lambda}{2} \|\mathbf{w}\|^2$$

$$F_{2,k}(\mathbf{w}) = \frac{1}{\sqrt{2k}} (|w[1] - w[2]| + |w[3] - w[4]| + \cdots + |w[k - 1] - w[k]|) + \frac{\lambda}{2} \|\mathbf{w}\|^2 \qquad (8)$$

For even $k \leq d$, and

$$F_{1,k}(\mathbf{w}) = \frac{1}{\sqrt{2}} |b - w[1]| + \frac{1}{\sqrt{2k}} (|w[2] - w[3]| + |w[4] - w[5]| + \cdots + |w[k-1] - w[k]|) + \frac{\lambda}{2} \|\mathbf{w}\|^2$$

$$F_{2,k}(\mathbf{w}) = \frac{1}{\sqrt{2k}} (|w[1] - w[2]| + |w[3] - w[4]| + \cdots + |w[k-2] - w[k-1]|) + \frac{\lambda}{2} \|\mathbf{w}\|^2$$

otherwise. Being a sum of convex functions, both local functions are convex, and in fact $\lambda$-strongly convex due to the $\frac{\lambda}{2} \|\mathbf{w}\|^2$ term. Furthermore, both function are $(1 + \lambda)$-Lipschitz continuous over the unit Euclidean ball. To see this, let $\partial(\cdot)$ denote the subdifferential operator and note that

$$\mathbf{g} \in \partial |b - w[1]| \implies \mathbf{g} \in \mathrm{conv}\{\boldsymbol{\sigma}_0, -\boldsymbol{\sigma}_0\}$$
$$\mathbf{g} \in \partial |w[l] - w[l+1]| \implies \mathbf{g} \in \mathrm{conv}\{\boldsymbol{\sigma}_l, -\boldsymbol{\sigma}_l\}$$

where

$$\boldsymbol{\sigma}_0 = (1, 0, \ldots, 0)$$
$$\boldsymbol{\sigma}_l = (0, \ldots, 0, \underbrace{1}_{l}, \underbrace{-1}_{l+1}, 0, \ldots, 0).$$

Assume for a moment that $\lambda = 0$, then by the linearity of the sub-differential operator that

$$\forall \mathbf{g} \in \partial F_{1,k}(\mathbf{w}), \ \|\mathbf{g}\| \leq \sqrt{\frac{1}{2} + \frac{k-1}{2k}} \leq 1$$

$$\forall \mathbf{g} \in \partial F_{2,k}(\mathbf{w}), \ \|\mathbf{g}\| \leq \sqrt{\frac{1}{2}}$$

which shows that, for $\lambda = 0$, both functions are 1-Lipschitz. For $\lambda > 0$, note that $\frac{\lambda}{2} \|\mathbf{w}\|^2$ is $\lambda$-Lipschitz over the unit ball and $\lambda$-strongly convex. Therefore, using the linearity of the sub-differential operator again, we see that both $F_i$ are $(1+\lambda)$-Lipschitz and $\lambda$-strongly convex functions over the unit ball.

Similar to the smooth case, the following lemma shows that, no matter how the subgradients are chosen, at each iteration at most one non-zero coordinate may be gained.

**Lemma 2.** *Suppose all the sets of feasible points satisfy $W_j \subseteq E_{T,d}$ for some $T \leq d - 1$. Then under assumption 1, right after the next communication round we have $W_j \subseteq E_{T+1,d}$.*

*Proof.* Recall that by Assumption 1 (modified for the non-differentiable case), each machine can compute new points $\mathbf{w}$ that satisfy the following for some $\gamma, \nu \geq 0$ such that $\gamma + \nu > 0$:

$$\gamma \mathbf{w} + \nu \mathbf{g}_{i,k}(\mathbf{w}) \in \mathrm{span}\Big\{\mathbf{w}' , \ \mathbf{g}_{i,k}(\mathbf{w}') , \ (\nabla^2 F_{i,k}(\mathbf{w}') + D)\mathbf{w}'' , \ (\nabla^2 F_{i,k}(\mathbf{w}') + D)^{-1}\mathbf{w}'' \ \Big|$$

$$\mathbf{w}', \mathbf{w}'' \in W_j , \ \mathbf{g}_{i,k}(\mathbf{w}') \in \partial F_{i,k}(\mathbf{w}') , \ D \text{ diagonal} , \ \nabla^2 F_{i,k}(\mathbf{w}') \text{ exists} , \ (\nabla^2 F_{i,k}(\mathbf{w}') + D)^{-1} \text{ exists}\Big\}.$$

We now analyze the state of the sets of feasible points prior to the next communication round. Assume that $T$ is an odd number, i.e., assume $T = 2p + 1$ for some $p \in \mathbb{N} \cup \{0\}$. We show that as long as no communication round has been executed, it must hold that $W_j \subseteq E_{T,d}$ for machines whose local function is $F_1$, and that $W_j \subseteq E_{T+1,d}$ for machines whose local function is $F_2$. The case where $T$ is even follows a similar line.

Let

$$E_{0,d} = \{\mathbf{0}\} \ , \ E_{T,d} = \mathrm{span}\{\mathbf{e}_{1,d}, \ldots, \mathbf{e}_{T,d}\}$$

where $\mathbf{e}_{i,d} \in \mathbb{R}^d$ denote the standard unit vectors. First, we prove the claim for machines whose local function is $F_{1,k}$. In which case, for any $\mathbf{w}', \mathbf{w}'' \in E_{2p+1,d}$, it holds that

$$\mathbf{g}_{1,k}(\mathbf{w}') \subseteq \mathrm{conv}\{\pm\boldsymbol{\sigma}_{2l} \mid l = 0, \ldots, p\} \subseteq E_{2p+1,d}$$
$$(\nabla^2 F_{1,k}(\mathbf{w}') + D)\mathbf{w}'' = (\lambda I + D) \mathbf{w}'' \subseteq E_{2p+1,d}$$
$$(\nabla^2 F_{1,k}(\mathbf{w}') + D)^{-1}\mathbf{w}'' = (\lambda I + D)^{-1} \mathbf{w}'' \subseteq E_{2p+1,d}$$

For any viable diagonal matrix D. Therefore, we have that the first point generated by machines which hold $F_{1,k}$ must satisfy

$$\gamma \mathbf{w} + \nu \mathbf{g}_{1,k}(\mathbf{w}) \in E_{2p+1,d} \tag{9}$$

for $\gamma, \nu$ as stated in Assumption (1). Note that, if $\nu = 0$ (which by assumption means that $\gamma > 0$) then clearly $\mathbf{w} \in E_{2p+1,d}$. As for $\nu \neq 0$, suppose by contradiction that $\mathbf{w} \notin E_{2p+1,d}$. That is, assume that there exists some $j > 2p + 1$ such that $w[j] \neq 0$. First, if the absolute value terms in $F_{1,k}$ do not involve $w[j]$, e.g., when $j = d$ and $d$ is even, we have $\mathbf{g}_{1,k}(\mathbf{w})[j] = \lambda w[j]$. In this case, by Eq. (9) we have

$$\gamma w[j] + \nu \lambda w[j] = (\gamma + \nu\lambda)w[j] = 0,$$

and since $\nu\lambda > 0$, this implies that $w[j] = 0$ – a contradiction! Thus, it remains to consider the cases of either odd $d$ or $j \neq d$. In both of these cases, $w[j]$ appears in one of the absolute value terms in $F_{1,k}$, either as $|w[j-1] - w[j]|$ or $|w[j] - w[j+1]|$ (depending on whether $j$ is odd or even).

Let $l > p$ be such that either $2l = j$ or $2l + 1 = j$, depending on the parity of $j$. We note that any valid subgradient must satisfy

$$\mathbf{g}_{1,k}(\mathbf{w})[2l] = \frac{2\alpha - 1}{\sqrt{2k}} + \lambda w[2l]$$

$$\mathbf{g}_{1,k}(\mathbf{w})[2l + 1] = \frac{1 - 2\alpha}{\sqrt{2k}} + \lambda w[2l + 1]$$

for some $\alpha \in [0, 1]$, such that if $w[2l] - w[2l + 1] \neq 0$ then

$$\operatorname{sgn}(w[2l] - w[2l + 1]) = \operatorname{sgn}\left(\frac{2\alpha - 1}{\sqrt{2k}}\right), \tag{10}$$

where $\operatorname{sgn}()$ is the sign function. Rearranging terms in Eq. (9) and using the facts that coordinates $2l, 2l + 1$ are always zero in $E_{2p+1,d}$, as well as $\gamma + \nu\lambda \geq \nu\lambda > 0$, we get

$$w[2l] = \frac{-\nu(2\alpha - 1)}{\sqrt{2k}(\gamma + \nu\lambda)} \tag{11}$$

$$w[2l + 1] = \frac{\nu(2\alpha - 1)}{\sqrt{2k}(\gamma + \nu\lambda)}$$

Therefore,

$$w[2l] - w[2l + 1] = \frac{-2\nu(2\alpha - 1)}{\sqrt{2k}(\gamma + \nu\lambda)} \tag{12}$$

which implies

$$\operatorname{sgn}(w[2l] - w[2l + 1]) = \operatorname{sgn}\left(\frac{-(2\alpha - 1)}{\sqrt{2k}}\right),$$

contradicting Eq. (10). Hence, we must have $w[2l] - w[2l + 1] = 0$, in which case Eq. (12) implies $\alpha = 1/2$. Thus, by Eq. (11)

$$w[2l] = w[2l + 1] = 0,$$

which contradicts the assumption that either $w[j]$ (and hence $w[2l]$ or $w[2l+1]$) is not zero. Thus, we have shown that $\mathbf{w} \in E_{2p+1,d}$, for the first point generated by machines holding $F_{1,k}$. Repeating the argument, we get that any point generated by those machines, in the absence of any communication rounds, is 'stuck' in $E_{2p+1,d}$.

We now turn to prove the claim for machines whose local function is $F_{2,k}$, using an almost identical argument, which we provide below for completeness. For these functions, we assume that initially $W_j \subseteq E_{2p+1,d}$, and will show any additional points computed locally by the machines must be in $E_{2p+2,d}$. We begin by noting that for any $\mathbf{w}', \mathbf{w}''$ in $E_{2p+2,d}$ (and in particular $E_{2p+1,d}$), it holds that

$$\mathbf{g}_{2,k}(\mathbf{w}') \subseteq \operatorname{conv}\{\pm\boldsymbol{\sigma}_{2l+1} \mid l = 0, \ldots, p\} \subseteq E_{2p+2,d}$$

$$(\nabla^2 F_{2,k}(\mathbf{w}') + D)\mathbf{w}'' = (\lambda I + D)\,\mathbf{w}'' \subseteq E_{2p+2,d}$$

$$(\nabla^2 F_{2,k}(\mathbf{w}') + D)^{-1}\mathbf{w}'' = (\lambda I + D)^{-1}\,\mathbf{w}'' \subseteq E_{2p+2,d}$$

For any viable diagonal matrix D. Therefore, we have that the first point generated by machines which hold $F_{2,k}$ must satisfy

$$\gamma \mathbf{w} + \nu \mathbf{g}_{2,k}(\mathbf{w}) \in E_{2p+2,d} \tag{13}$$

for $\gamma, \nu$ as stated in the assumption. Note that, if $\nu = 0$ then clearly $\mathbf{w} \in E_{2p+2,d}$. As for $\nu \neq 0$, suppose by contradiction that $\mathbf{w} \notin E_{2p+2,d}$. That is, assume that there exists some $j > 2p + 2$ such that $w[j] \neq 0$. First, if the absolute value terms in $F_{2,k}$ do not involve $w[j]$, e.g., when $j = d$ and $d$ is odd, we have $\mathbf{g}_{2,k}(\mathbf{w})[j] = \lambda w[j]$. In this case, by Eq. (13) we have

$$\gamma w[j] + \nu \lambda w[j] = (\gamma + \nu \lambda) w[j] = 0,$$

and since $\nu \lambda > 0$, this implies that $w[j] = 0$ – a contradiction! Thus, it remains to consider the cases of either even $d$ or $j \neq d$. In both of these cases, $w[j]$ appears in one of the absolute value terms in $F_{2,k}$, either as $|w[j-1] - w[j]|$ or $|w[j] - w[j+1]|$ (depending on whether $j$ is odd or even).

Let $l > p$ be such that either $2l + 1 = j$ or $2l + 2 = j$, depending on the parity of $j$. We note that any valid subgradient must satisfy Any valid subgradient must satisfy

$$\mathbf{g}_{2,k}(\mathbf{w})[2l+1] = \frac{2\alpha - 1}{\sqrt{2k}} + \lambda w[2l+1]$$

$$\mathbf{g}_{2,k}(\mathbf{w})[2l+2] = \frac{1 - 2\alpha}{\sqrt{2k}} + \lambda w[2l+2]$$

for some $\alpha \in [0, 1]$, such that if $w[2l+1] - w[2l+2] \neq 0$ then

$$\text{sgn}\left(w[2l+1] - w[2l+2]\right) = \text{sgn}\left(\frac{2\alpha - 1}{\sqrt{2k}}\right) \tag{14}$$

Rearranging terms in Eq. (13) and using the fact that $\gamma + \nu \lambda \geq \nu \lambda > 0$, we get

$$w[2l+1] = \frac{-\nu(2\alpha - 1)}{\sqrt{2k}(\gamma + \nu \lambda)} \tag{15}$$

$$w[2l+2] = \frac{\nu(2\alpha - 1)}{\sqrt{2k}(\gamma + \nu \lambda)}$$

Therefore,

$$w[2l+1] - w[2l+2] = \frac{-2\nu(2\alpha - 1)}{\sqrt{2k}(\gamma + \nu \lambda)} \tag{16}$$

which implies

$$\text{sgn}\left(w[2l+1] - w[2l+2]\right) = \text{sgn}\left(\frac{-(2\alpha - 1)}{\sqrt{2k}}\right)$$

a contradiction to Eq. (14). Hence, we must have $w[2l+1] - w[2l+2] = 0$, in which case Eq. (16) implies $\alpha = 1/2$. Thus, by Eq. (15)

$$w[2l+1] = w[2l+2] = 0$$

which contradicts our assumption that $w[j]$ (and hence either $w[2l+1]$ or $w[2l+2]$ is not zero). Thus, we have shown that $\mathbf{w} \in E_{2p+2,d}$. As before, repeating the argument together with the assumption that $W_j \subseteq E_{2p+2,d}$ shows that, in the absence of any communication rounds, all the machines whose local function is $F_{2,k}$ are 'stuck' in $E_{2p+2,d}$. Therefore, before the next communication round, $W_j \subseteq E_{2p+2,d}$ for all machines $j$ holding $F_{2,j}$. Moreover, as shown earlier, $W_j \subseteq E_{2p+1,d}$ for all machines holding $F_{1,k}$. Therefore, after the next communication round, $W_j \subseteq E_{2p+2,d}$ for any machine $j$. $\qquad\square$

Repeatedly applying Lemma 2, we get the following corollary:

**Corollary 2.** *Under assumption 1, after $T \leq d - 1$ communication rounds we have*

$$W_j \subseteq E_{T+1}, \quad j \in \{1, \ldots, m\}$$

With this corollary in hand, we now turn to establish the main result, namely, bounding from below the optimality of points in $W_j$ after $T$ communication rounds. Choosing the dimension $d$ such that $T \leq d - 2$, we employ the local functions defined in Eq. (8) with $k = T + 2$. In which case, the average function is

$$F(\mathbf{w}) = \frac{1}{2}F_{1,T+2}(\mathbf{w}) + \frac{1}{2}F_{2,T+2}(\mathbf{w}) = \frac{1}{2\sqrt{2}}|b - w[1]| + \frac{1}{2\sqrt{2(T+2)}}\sum_{i=1}^{T+1}|w[i] - w[i+1]| + \frac{\lambda}{2}\|\mathbf{w}\|^2$$

The key ingredient in deriving the lower bound is Corollary (2), according to which after $T$ communication rounds, all but the first $T + 1$ coordinates must be zero, in particular $w[T + 2] = 0$. Using this and the triangle inequality, we have

$$F(\mathbf{w}) \geq \frac{1}{2\sqrt{2}}|b - w[1]| + \frac{1}{2\sqrt{2(T+2)}}|w[1] - w[T+2]| + \frac{\lambda}{2}\|\mathbf{w}\|^2$$

$$= \frac{1}{2\sqrt{2}}|b - w[1]| + \frac{1}{2\sqrt{2(T+2)}}|w[1]| + \frac{\lambda w[1]^2}{2}$$

for all $\mathbf{w}$ in $W_j$. Therefore, we can lower bound the objective value of the algorithm's output by

$$\min_{w \in \mathbb{R}} \left( \frac{1}{2\sqrt{2}}|b - w| + \frac{1}{2\sqrt{2(T+2)}}|w| + \frac{\lambda w^2}{2} \right)$$

On the flip side, the minimal value of $F(\mathbf{w})$ over the unit Euclidean ball can be upper bounded by $F(\mathbf{w}_b)$ for some $b \leq \frac{1}{\sqrt{T+2}}$, where

$$\mathbf{w}_b = (\underbrace{b, \ldots, b}_{T+2 \text{ times}}, 0, \ldots, 0)$$

Putting both bounds together yields,

$$\min_{\mathbf{w} \in W_j} F(\mathbf{w}) - \min_{\|\mathbf{w}\| \leq 1} F(\mathbf{w}) \geq \min_{\mathbf{w} \in W_j} F(\mathbf{w}) - F(\mathbf{w}_b)$$

$$\geq \min_{w \in \mathbb{R}} \left( \frac{1}{2\sqrt{2}}|b - w| + \frac{1}{2\sqrt{2(T+2)}}|w| + \frac{\lambda w^2}{2} \right) - \frac{\lambda(T+2)b^2}{2}$$

$$\tag{17}$$

Assuming $T \geq \frac{1}{2\lambda} - 2$ (so that $\lambda \geq \frac{1}{2(T+2)}$), we take

$$b = \frac{1}{2\lambda(T+2)\sqrt{2(T+2)}}$$

(note again that $\mathbf{w}_b$ is indeed in the unit ball for this regime of $\lambda$ and $T$). In this case, the minimal $w$ in Eq. (17) is $\frac{1}{2\lambda(T+2)\sqrt{2(T+2)}}$, so we get a suboptimality lower bound of

$$\left( 0 + \frac{1}{2\sqrt{2(T+2)}} \left| \frac{1}{2\lambda(T+2)\sqrt{2(T+2)}} \right| + \frac{1}{8\lambda\left((T+2)\sqrt{2(T+2)}^2\right)} \right) - \frac{1}{16\lambda(T+2)^2}$$

$$\geq \left( \frac{1}{8\lambda(T+2)^2} + 0 \right) - \frac{1}{16\lambda(T+2)^2}$$

$$= \frac{1}{16\lambda(T+2)^2} \tag{18}$$

This bound holds in particular for any $T \geq \left\lceil \frac{1}{2\lambda} - 2 \right\rceil$. If the number of communication rounds $T$ is less than $\left\lceil \frac{1}{2\lambda} - 2 \right\rceil$, then clearly we cannot do better than with $\left\lceil \frac{1}{2\lambda} - 2 \right\rceil$ communication rounds. Therefore, for any number of communication rounds $T$, the suboptimality is at least

$$\min \left\{ \frac{1}{16\lambda\left(\left\lceil \frac{1}{2\lambda} - 2 \right\rceil + 2\right)^2}, \frac{1}{16\lambda(T+2)^2} \right\}$$

Therefore, for any $\epsilon \in \left(0, \frac{1}{16\lambda\left(\lceil\frac{1}{2\lambda}-2\rceil+2\right)^2}\right]$, we would need at least $T \geq \sqrt{\frac{1}{16\lambda\epsilon}} - 2$ communication rounds to get an $\epsilon$-suboptimal solution. This implies the theorem statement for $\lambda$-strongly convex functions.

Finally, we treat the case where the local functions are not required to be strongly convex. In this setting, for proving a lower bound, we can use the same construction as in Eq. (8), where we are free to choose any $\lambda$. In particular, let us choose $\lambda = \frac{1}{2(T+2)}$, and apply the lower bound derived above (note that in this case the condition $T \geq \frac{1}{2\lambda} - 2$ trivially holds). Plugging in it into (18), we establish that for any number of communication rounds $T$, the suboptimality is at least

$$\frac{1}{8(T+2)}.$$

Considering how large $T$ must be to make this smaller than some $\epsilon$, we get that $T$ must be at least $\frac{1}{8\epsilon} - 2$.

### A.3 Proof of Thm. 3

As usual, we construct two functions $F_1, F_2$, and provide $F_1$ to $m/2$ of the machines, and $F_2$ to the other $m/2$ machines, in some arbitrary order, such that the machine designated to provide the output receives $F_2$. Note that the average function $F$ is simply $\frac{1}{2}(F_1(\mathbf{w}) + F_2(\mathbf{w}))$.

Let $c$ be a certain positive numerical constant (whose value corresponds to $c$ in Lemma 6 below). Given some symmetric $M \in \{-1, +1\}^{d \times d}$, where $\|M\| \leq c\sqrt{d}$, and $j \in \{\lceil d/2\rceil, \ldots, d\}$, define

$$F_1(\mathbf{w}) = 3\lambda\mathbf{w}^\top \left(\left(I + \frac{1}{2c\sqrt{d}}M\right)^{-1} - \frac{1}{2}I\right)\mathbf{w}$$

$$F_2(\mathbf{w}) = \frac{3\lambda}{2}\|\mathbf{w}\|^2 - \delta\mathbf{e}_j,$$

The average $F$ of $F_1, F_2$ equals

$$F(\mathbf{w}) = \frac{1}{2}(F_1(\mathbf{w}) + F_2(\mathbf{w})) = \frac{3\lambda}{2}\mathbf{w}^\top\left(I + \frac{1}{2c\sqrt{d}}M\right)^{-1}\mathbf{w} - \frac{\delta}{2}\mathbf{e}_j,$$

with an optimum at

$$\mathbf{w}^* = \frac{\delta}{6\lambda}\left(I + \frac{1}{2c\sqrt{d}}M\right)\mathbf{e}_j.$$

The following lemma establishes that the functions satisfy the strong convexity, smoothness and relatedness requirements of the theorem. The proof also establishes that the inverse in the definition of $F_1$ indeed exists.

**Lemma 3.** $F_1$ and $F_2$ are $\lambda$ strongly-convex, $9\lambda$ smooth, and $\delta$-related.

*Proof.* The Hessian of $F_2$ is $3\lambda I$, which implies that $F_2$ is $3\lambda$ smooth and strongly convex (and in particular, $\lambda$-strongly convex). As to $F_1$, note that since $\|M\| \leq c\sqrt{d}$, then

$$\left\|\frac{1}{2c\sqrt{d}}M\right\| \leq \frac{1}{2},$$

The fact that the spectral radius and spectral norm of symmetric matrices coincide implies that the eigenvalues of the matrix $I + \frac{1}{2c\sqrt{d}}M$ lie between $1 - \frac{1}{2} = \frac{1}{2}$ and $1 + \frac{1}{2} = \frac{3}{2}$. Thus, all the eigenvalues are strictly positive, hence the matrix is indeed invertible as in the definition of $F_1$. Moreover, the eigenvalues of the inverse lie in $\left[\frac{1}{3/2}, \frac{1}{1/2}\right] = \left[\frac{2}{3}, 2\right]$, and therefore those of $3\lambda\left(\left(I + \frac{1}{2c\sqrt{d}}M\right)^{-1} - \frac{1}{2}I\right)$ lie in $\left[3\lambda\left(\frac{2}{3} - \frac{1}{2}\right), 3\lambda\left(2 - \frac{1}{2}\right)\right] = \left[\frac{\lambda}{2}, \frac{9\lambda}{2}\right]$. Thus, the spectrum of the Hessian of $F_1$ lie in $[\lambda, 9\lambda]$, which implies that $F_1$ is $\lambda$-strongly convex and $9\lambda$ smooth.

To show $\delta$-relatedness, the only non-trivial part is upper-bounding the norm of the difference of the quadratic terms, which equals the following:

$$\left\| 3\lambda\left(\left(I+\frac{1}{2c\sqrt{d}}M\right)^{-1}-\frac{1}{2}I\right)-\frac{3\lambda}{2}I\right\|$$

$$= 3\lambda\left\|\left(I+\frac{1}{2c\sqrt{d}}M\right)^{-1}-I\right\|. \tag{19}$$

Since $\|M\| \leq c\sqrt{d}$, the eigenvalues of $\left(I+\frac{1}{2c\sqrt{d}}M\right)^{-1}-I$ lie between $\frac{1}{1+1/2}-1 = -\frac{1}{3}$ and $\frac{1}{1-1/2}-1 = 1$, which implies that Eq. (19) can be upper bounded by $3\lambda \leq \delta$. $\qquad\square$

The next lemma proves the second part of the theorem, namely an upper bound on the suboptimality of any local function optimizer.

**Lemma 4.** *For any $\hat{\mathbf{w}}_j = \arg\min_{\mathbf{w}\in\mathbb{R}^d} F_j(\mathbf{w})$, it holds that $F(\hat{\mathbf{w}}_j) - \min_{\mathbf{w}\in\mathbb{R}^d} F(\mathbf{w}) \leq c\delta^2/\lambda$ for some numerical positive constant $c$.*

*Proof.* The optimum of any quadratic and strongly-convex function $\mathbf{w}^\top A\mathbf{w} + \mathbf{b}^\top\mathbf{w} + c$ equals $\frac{1}{2}A^{-1}\mathbf{b}$. Therefore, if $\mathbf{w}^*$ is the optimizer of $F$, and we denote the parameters of $F$ and $F_j$ by $A, \mathbf{b}, c$ and $A_j, \mathbf{b}_j, c_j$ respectively, then

$$\|\hat{\mathbf{w}}_j - \mathbf{w}^*\| = \frac{1}{2}\left\|A_j^{-1}\mathbf{b}_j - A^{-1}\mathbf{b}\right\|$$

$$= \frac{1}{2}\left\|A_j^{-1}\mathbf{b}_j - A^{-1}\mathbf{b}_j + A^{-1}\mathbf{b}_j - A^{-1}\mathbf{b}\right\|$$

$$\leq \frac{1}{2}\left(\left\|\left(A_j^{-1}-A^{-1}\right)\mathbf{b}_j\right\| + \left\|A^{-1}\left(\mathbf{b}_j-\mathbf{b}\right)\right\|\right)$$

$$\leq \frac{1}{2}\left(\left\|A_j^{-1}-A^{-1}\right\|\|\mathbf{b}_j\| + \left\|A^{-1}\right\|\|\mathbf{b}_j-\mathbf{b}\|\right).$$

By definition of $F_1, F_2$ and the average function $F$, this is at most

$$\frac{1}{2}\left(\left\|A_j^{-1}-A^{-1}\right\|\delta + \left\|A^{-1}\right\|\frac{\delta}{2}\right). \tag{20}$$

In Lemma 3, we showed that $F_1, F_2$ are $\lambda$-strongly convex and $9\lambda$ smooth, which implies that the eigenvalues of $A_j$ as well as $A$ lie in $\left[\frac{\lambda}{2}, \frac{9\lambda}{2}\right]$. Therefore, the eigenvalues of $A_j^{-1}$ and $A^{-1}$ lie in $\left[\frac{2}{9\lambda}, \frac{2}{\lambda}\right]$, so $\|A^{-1}\| \leq \frac{2}{\lambda}$ and $\|A_j^{-1}-A^{-1}\| \leq \frac{2}{\lambda}$. Substituting this back into Eq. (20), we get

$$\|\hat{\mathbf{w}}_j - \mathbf{w}^*\| \leq \frac{1}{\lambda}\left(\delta + \frac{\delta}{2}\right) = \frac{3\delta}{2\lambda}.$$

Finally, since $F$ is $9\lambda$-smooth, and its minimizer is $\mathbf{w}^*$,

$$F(\hat{\mathbf{w}}_j) - F(\mathbf{w}^*) \leq \frac{9\lambda}{2}\|\hat{\mathbf{w}}_j - \mathbf{w}^*\|^2 \leq \frac{9\lambda}{2}\left(\frac{3\delta}{2\lambda}\right)^2,$$

which equals $81\delta^2/8\lambda$ as required. $\qquad\square$

We now turn to derive the lower bound in the theorem statement. As discussed earlier, the intuition is that the optimal point $\mathbf{w}^*$ is a function of the $j$-th column of $M$, so the machines holding $F_1$ must broadcast enough information on $M$ to the designated machine producing the algorithm's output (the machine, by construction, holds $F_2$, and hence knows $j$ but not $M$). As long as the communication budget is smaller than the size of $M$, this will be difficult to achieve. This intuition is formalized in the following lemma, which is based on information-theoretic tools:

**Lemma 5.** *For any dimension $d \geq c$ (where $c$ is the same constant as in Lemma 6 and the definition of $F_1$), and for any (possibly randomized) 1-round algorithm using at most $d^2/128$ bits of communication, there exists a valid choice of $M, j$ for the functions $F_1, F_2$ defined above, such that the vector $\hat{\mathbf{w}}$ returned by the algorithm satisfies*

$$\mathbb{E}\left[\|\hat{\mathbf{w}} - \mathbf{w}^*\|^2\right] \geq c'\left(\frac{\delta}{\lambda}\right)^2,$$

*where the expectation is over the algorithm's randomness, and $c'$ is a positive numerical constant.*

Using the lemma and the $\lambda$-strong convexity of $F_1, F_2$ (and hence their average $F$),

$$\mathbb{E}[F(\hat{\mathbf{w}}) - F(\mathbf{w}^*)] \geq \frac{\lambda}{2}\mathbb{E}[\|\mathbf{w} - \mathbf{w}^*\|^2] \geq \frac{c'}{2}\frac{\delta^2}{\lambda},$$

hence proving the theorem.

It now remains to prove Lemma 5:

*Proof of Lemma 5.* By definition of $\mathbf{w}^*$, we have that the $j$-th column of $M$, designated as $M_j$, satisfies

$$M_j = 2c\sqrt{d}\left(\frac{6\lambda}{\delta}\mathbf{w}^* - \mathbf{e}_j\right).$$

Given the predictor $\hat{\mathbf{w}}$ returned by the algorithm, define

$$\hat{M}_j = 2c\sqrt{d}\left(\frac{6\lambda}{\delta}\hat{\mathbf{w}} - \mathbf{e}_j\right).$$

This can be thought of as the algorithm's 'estimate' of the $j$-th column of $M$, based on the returned predictor.

Define $[w] = \min\{1, \max\{-1, w\}\}$ as the clipping operation of a scalar $w$ to $[-1, +1]$, and for a vector $\mathbf{w} = (w_1, \ldots, w_d)$, define $[\mathbf{w}] = ([w_1], [w_2], \ldots, [w_d])$. By the expressions for $M_j, \hat{M}_j$ above, we have

$$\left\|[\hat{M}_j - M_j]\right\|^2 = \left\|\left[2c\sqrt{d}\frac{6\lambda}{\delta}(\hat{\mathbf{w}} - \mathbf{w}^*)\right]\right\|^2 = \sum_{i=1}^{d}\left[\frac{12c\lambda\sqrt{d}}{\delta}(\hat{w}_i - w_i^*)\right]^2$$

$$\leq \left(\frac{12c\lambda\sqrt{d}}{\delta}\right)^2\sum_{i=1}^{d}(\hat{w}_i - w_i^*)^2,$$

which implies that

$$\|\hat{\mathbf{w}} - \mathbf{w}^*\|^2 \geq \left(\frac{\delta}{12c\lambda\sqrt{d}}\right)^2\left\|[\hat{M}_j - M_j]\right\|^2, \tag{21}$$

To get the lemma statement, it is enough to show that for some $M, j$, one can lower bound $\mathbb{E}\left[\left\|[\hat{M}_j - M_j]\right\|^2\right]$ (where the expectation is over the algorithm's randomness) by some constant multiple of $d$.

Below, we will prove that if $M$ (in the definition of $F_1$) is chosen uniformly at random from all $\{-1, +1\}$-valued $d \times d$ symmetric matrices, and $j$ (in the definition of $F_2$) is chosen uniformly at random from $\{\lceil d/2 \rceil, \ldots, d\}$, then for any deterministic algorithm,

$$\mathbb{E}_{M,j}\left[\left\|[\hat{M}_j - M_j]\right\|^2\right] \geq \frac{d}{8} \tag{22}$$

Let us first show how this can be used to prove the lemma. To do so, we will need the following lemma on the concentration of the spectral norm of random symmetric matrices.

**Lemma 6** ([21], Corollary 2.3.6). *There exist positive numerical constants $c, c'$, such that if $M$ is a $d \times d$ symmetric matrix, where each entry $M_{j,i}, j \geq i$ is chosen independently and uniformly from $\{-1, +1\}$, and $d \geq c$, then $\Pr(\|M\| > c\sqrt{d}) \leq c\exp(-c'd)$.*

First, we note that the expectation in Eq. (22) is over all symmetric $\{-1, +1\}$-valued matrices, including those whose spectral norm may be larger than $c\sqrt{d}$. However, by Lemma 6, $\Pr(\|M\| > c\sqrt{d}) \leq c\exp(-c'd)$ for some absolute constant $c'$. Letting $E$ be the event that $\|M\| > c\sqrt{d}$, and noting that $\|[\mathbf{w}]\|^2 \leq d$ for any vector $\mathbf{w}$, we have

$$\mathbb{E}\left[\left\|[\hat{M}_j - M_j]\right\|^2\right] = \mathbb{E}\left[\left\|[\hat{M}_j - M_j]\right\|^2 \Big| E\right]\Pr(E) + \mathbb{E}\left[\left\|[\hat{M}_j - M_j]\right\|^2 \Big| \neg E\right]\Pr(\neg E)$$

$$\leq d\Pr(E) + \mathbb{E}\left[\left\|[\hat{M}_j - M_j]\right\|^2 \Big| \neg E\right]$$

$$\leq cd\exp(-c'd) + \mathbb{E}\left[\left\|[\hat{M}_j - M_j]\right\|^2 \Big| \neg E\right].$$

Plugging back into Eq. (22), we get that

$$\mathbb{E}\left[\left\|[\hat{M}_j - M_j]\right\|^2 \Big| \neg E\right] \geq \frac{d}{8} - cd\exp(-c_1 d),$$

which is at least $d/16$ for any $d$ larger than some constant. Combining with Eq. (21), we get

$$\mathbb{E}\left[\|\hat{\mathbf{w}} - \mathbf{w}^*\|^2 \Big| \neg E\right] \geq \frac{1}{16}\left(\frac{\delta}{12c\lambda}\right)^2.$$

This inequality implies that for any deterministic algorithm, in expectation over the random draw of $j$ and a $\{-1, +1\}$-valued matrix $M$ with spectral norm at most $c\sqrt{d}$, $\|\hat{\mathbf{w}} - \mathbf{w}^*\|^2$ will be at least $c'\left(\frac{\delta}{\lambda}\right)^2$ for some suitable constant $c'$. By Yao's minimax principle, this implies that for any (possibly randomized) algorithm, there will be some deterministic choice of $M, j$ such that $\|M\| \leq c\sqrt{d}$, and for which

$$\mathbb{E}\left[\|\hat{\mathbf{w}} - \mathbf{w}^*\|^2\right] \geq c'\left(\frac{\delta}{\lambda}\right)^2$$

(in expectation over the algorithm's randomness), yielding the lemma's statement.

It now remains to prove Eq. (22), assuming $j$ is chosen uniformly at random from $\{\lceil d/2 \rceil, \ldots, d\}$, and $M$ is chosen at random (i.e. each entry at or above the main diagonal is chosen independently and uniformly from $\{-1, +1\}$). Roughly speaking, the proof idea is to reduce this to an upper bound on how much information the machines holding $M$ can send on $M$'s entries (and more particularly, on the entries in the upper-right quadrant of $M$). Since this quadrant is composed of $\Theta(d^2)$ random variables, and the machines can send much less than $d^2$ bits, this information is necessarily restricted.

Let $\Pr(\cdot)$ denote probability with respect to the random choice of $M, j$, and let $\Pr_j(\cdot)$ denote probability conditioned on the choice of $j$. Recalling that any entry $M_{j,i}$ in the $j$-th column has values in $\{-1, +1\}$, it follows that either $M_{j,i}$ has the same sign as $\hat{M}_{j,i}$, or that $([M_{j,i} - \hat{M}_{j,i}])^2$ is at least

1. Therefore, we have the following:

$$
\mathbb{E}\left[\left\|[M_j - \hat{M}_j]\right\|^2\right] = \sum_{i=1}^{d} \mathbb{E}[([M_{j,i} - \hat{M}_{j,i}])^2] \geq \sum_{i=1}^{\lceil d/2 \rceil} \mathbb{E}[([M_{j,i} - \hat{M}_{j,i}])^2]
$$

$$
\geq \sum_{i=1}^{\lceil d/2 \rceil} \mathbb{E}\left[([M_{j,i} - \hat{M}_{j,i}])^2 \Big| M_{j,i}\hat{M}_{j,i} \leq 0\right] \Pr\left(M_{j,i}\hat{M}_{j,i} \leq 0\right) + 0
$$

$$
\geq \sum_{i=1}^{\lceil d/2 \rceil} \Pr\left(M_{j,i}\hat{M}_{j,i} \leq 0\right) = \sum_{i=1}^{\lceil d/2 \rceil} \left(\frac{1}{1 + \lfloor d/2 \rfloor} \sum_{j=\lceil d/2 \rceil}^{d} \Pr_j\left(M_{j,i}\hat{M}_{j,i} \leq 0\right)\right)
$$

$$
= \frac{1}{1 + \lfloor d/2 \rfloor} \sum_{i=1}^{\lceil d/2 \rceil} \sum_{j=\lceil d/2 \rceil}^{d} \left(\frac{1}{2}\Pr_j\left(\hat{M}_{j,i} \leq 0 | M_{j,i} > 0\right) + \frac{1}{2}\Pr_j\left(\hat{M}_{j,i} \geq 0 | M_{j,i} < 0\right)\right)
$$

$$
\geq \frac{1/2}{1 + \lfloor d/2 \rfloor} \sum_{i=1}^{\lceil d/2 \rceil} \sum_{j=\lceil d/2 \rceil}^{d} \left(1 - \left(\Pr_j\left(\hat{M}_{j,i} \geq 0 | M_{j,i} > 0\right) - \Pr_j\left(\hat{M}_{j,i} \geq 0 | M_{j,i} < 0\right)\right)\right)
$$

$$
\geq \frac{1/2}{1 + \lfloor d/2 \rfloor} \sum_{i=1}^{\lceil d/2 \rceil} \sum_{j=\lceil d/2 \rceil}^{d} \left(1 - \left|\Pr_j\left(\hat{M}_{j,i} \geq 0 | M_{j,i} < 0\right) - \Pr_j\left(\hat{M}_{j,i} \geq 0 | M_{j,i} > 0\right)\right|\right)
$$

$$
= \frac{\lceil d/2 \rceil}{2} - \frac{1/2}{1 + \lfloor d/2 \rfloor} \sum_{i=1}^{\lceil d/2 \rceil} \sum_{j=\lceil d/2 \rceil}^{d} \left|\Pr_j\left(\hat{M}_{j,i} \geq 0 | M_{j,i} < 0\right) - \Pr_j\left(\hat{M}_{j,i} \geq 0 | M_{j,i} > 0\right)\right|.
$$

$$(23)$$

Let $S$ be the vector of bits broadcasted by the machines holding $F_1$, and received by the machine designated with providing the output (recalling that it only holds $F_2$). Note that conditioned on $S$ and $j$, the algorithm's output (and hence $\hat{M}_{j,i}$) is independent of $M$. Therefore, we have

$$
\left|\Pr_j\left(\hat{M}_{j,i} \geq 0 | M_{j,i} < 0\right) - \Pr_j\left(\hat{M}_{j,i} \geq 0 | M_{j,i} > 0\right)\right|
$$

$$
= \left|\sum_S \Pr_j\left(\hat{M}_{j,i} \geq 0 | S, M_{j,i} < 0\right) \Pr(S | M_{j,i} < 0) - \sum_S \Pr_j\left(\hat{M}_{j,i} \geq 0 | S, M_{j,i} > 0\right) \Pr(S | M_{j,i} > 0)\right|
$$

$$
= \left|\sum_S \Pr_j\left(\hat{M}_{j,i} \geq 0 | S\right) \Pr(S | M_{j,i} < 0) - \sum_S \Pr_j\left(\hat{M}_{j,i} \geq 0 | S\right) \Pr(S | M_{j,i} > 0)\right|
$$

$$
\leq \sum_S \left|\Pr_j\left(\hat{M}_{j,i} \geq 0 | S\right)\left(\Pr(S | M_{j,i} < 0) - \Pr(S | M_{j,i} > 0)\right)\right|
$$

$$
\leq \sum_S \left|\Pr_j(S | M_{j,i} < 0) - \Pr_j(S | M_{j,i} > 0)\right|
$$

$$
\leq \sum_S \left|\Pr_j(S | M_{j,i} < 0) - \Pr_j(S)\right| + \sum_S \left|\Pr_j(S | M_{j,i} > 0) - \Pr_j(S)\right|.
$$

Since $S$ is sent by the machines holding $F_1$ (and not $F_2$), it is independent of $j$. Therefore, we can write the above as

$$
\sum_S \left|\Pr(S | M_{j,i} < 0) - \Pr(S)\right| + \sum_S \left|\Pr(S | M_{j,i} > 0) - \Pr(S)\right|
$$

where $j$ in the conditioning is a fixed index. Using Pinsker's inequality, we can upper bound the above by

$$
\sqrt{2D_{kl}\left(p(S | M_{j,i} < 0) \| p(S)\right)} + \sqrt{2D_{kl}\left(p(S | M_{j,i} > 0) \| p(S)\right)}
$$

where $p$ is the probability distribution of $S$, and $D_{kl}$ is the Kullback-Leibler divergence. By the elementary inequality $\sqrt{a} + \sqrt{b} \leq \sqrt{2(a + b)}$ for all non-negative $a, b$, we can upper bound the

above by

$$\sqrt{4\left(D_{kl}\left(p(S|M_{j,i}<0)||p(S)\right)+D_{kl}\left(p(S|M_{j,i}>0)||p(S)\right)\right)}$$

$$=\sqrt{8}\sqrt{\frac{1}{2}\left(D_{kl}\left(p(S|M_{j,i}<0)||p(S)\right)+D_{kl}\left(p(S|M_{j,i}>0)||p(S)\right)\right)}.$$

Using the fact that $M_{j,i}$ (for some fixed $j,i$) is uniformly distributed in $\{-1,+1\}$, and that the mutual information $I(X;Y)$ between random variables $X,Y$ equals $\mathbb{E}_Y\left[D_{kl}(p(X|Y=y)||p(X))\right]$, the above equals

$$\sqrt{8}\sqrt{I(S;M_{j,i})}.$$

Recalling that this is an upper bound on $\left|\Pr_j\left(\hat{M}_{j,i}\geq0|M_{j,i}<0\right)-\Pr_j\left(\hat{M}_{j,i}\geq0|M_{j,i}>0\right)\right|$, we have

$$\frac{1/2}{1+\lfloor d/2\rfloor}\sum_{i=1}^{\lceil d/2\rceil}\sum_{j=\lceil d/2\rceil}^{d}\left|\Pr_j\left(\hat{M}_{j,i}\geq0|M_{j,i}<0\right)-\Pr_j\left(\hat{M}_{j,i}\geq0|M_{j,i}>0\right)\right|$$

$$\leq\frac{\sqrt{2}}{1+\lfloor d/2\rfloor}\sum_{i=1}^{\lceil d/2\rceil}\sum_{j=\lceil d/2\rceil}^{d}\sqrt{I(S;M_{j,i})}=\sqrt{2}\lceil d/2\rceil\frac{1}{\lceil d/2\rceil\left(1+\lfloor d/2\rfloor\right)}\sum_{i=1}^{\lceil d/2\rceil}\sum_{j=\lceil d/2\rceil}^{d}\sqrt{I(S;M_{j,i})}$$

$$\leq\sqrt{2}\lceil d/2\rceil\sqrt{\frac{1}{\lceil d/2\rceil\left(1+\lfloor d/2\rfloor\right)}\sum_{i=1}^{\lceil d/2\rceil}\sum_{j=\lceil d/2\rceil}^{d}I(S;M_{j,i})}\qquad(24)$$

where the last step is by Jensen's inequality (i.e. the average of square roots is upper bounded by the square root of the average). The expression in the square root equals the average mutual information between a random variable $S$ (composed of at most $d^2/128$ bits), and $\lceil d/2\rceil\left(1+\lfloor d/2\rfloor\right)$ binary random variables $M_{j,i}$, where $i\in\{1,\ldots,\lceil d/2\rceil\},j\in\{\lceil d/2\rceil,\ldots,d\}$, which are all independent by construction. By Lemma 6 in [18], it is at most $(d^2/128)/\left(\lceil d/2\rceil\left(1+\lfloor d/2\rfloor\right)\right)\leq1/32$, so we have

$$\sqrt{2}\lceil d/2\rceil\sqrt{\frac{1}{\lceil d/2\rceil\left(1+\lfloor d/2\rfloor\right)}\sum_{i=1}^{\lceil d/2\rceil}\sum_{j=\lceil d/2\rceil}^{d}I(S;M_{j,i})}\leq\sqrt{2}\lceil d/2\rceil\sqrt{\frac{1}{32}}=\frac{\lceil d/2\rceil}{4}.$$

Recalling this is an upper bound on Eq. (24), which is the second term in Eq. (23), we get that

$$\mathbb{E}_{M,j}\left[[M_j-\hat{M}_j]^2\right]\geq\frac{\lceil d/2\rceil}{2}-\frac{\lceil d/2\rceil}{4}=\frac{\lceil d/2\rceil}{4}\geq\frac{d}{8},$$

hence justifying Eq. (22). $\qquad\square$