[Reviews · NeurIPS 2015]

Submitted by Assigned_Reviewer_1

The major limitation of distributed optimization is communication cost of local parameters. So it is theoretically interesting to understand if it is possible to reduce this cost, either in terms of rounds of communication or the number of variables of each round of communication.

The proved theorems show the communication rounds scale polynomially with 1/error and 1/lambda for lambda strongly convex functions. This result assumes local solution is based on gradient (can be second order) descent. This is reasonably strong given that it covers a wide range of algorithms.

Given the big impact on distributed optimization of large scale learning, I recommend this paper to be accepted.
Summary: This paper proves theoretical communication rounds of distributed convex optimization problem with proper assumptions of each local optimization algorithm.

Submitted by Assigned_Reviewer_2

The paper gives lower bounds for the number of rounds needed for optimizing convex functions distributedly. In the main result of the paper (Theorem 1), it is assumed that we are optimizing a sum of delta-related and \lambda functions with a certain class of algorithms (which can basically take gradient, use local hessein to do preconditioning, and other related operations). It is shown that number of rounds for this task is at least \sqrt{\detla/\lambda}. These are matched by DiSCO under quadratic case. The class of functions concerned here contains the (probably the most) interesting situation, random partition of a set of functions, that people care about, though the former is a strict superset of the later.

The paper also shows lower bounds for the case when the local functions can be arbitrary, and the case when only one round of communication is allowed.

The main proof technique is similar in spirit to the query complexity of first order method. Essentially the constructed hard instances is such that starting from initial iterate 0, every round of communication can only provide information for one additional dimension and therefore without enough rounds, the iterate can only be in the span of the first several dimensions and dooms to have large error.

In summary, the paper has nice results and is well written. I wouldn't say that results are very strong due to the restrictions to the algorithms and the large class of functions concerned. However, proving lower bounds for rounds of communications are typically

hard (e.g. for the typical boolean function and two player case), and therefore restriction to a class of algorithms should be not be a problem.

Finally, there are some minor issues that I hope the author could clarify/revise/strengthen:

1. the argument seems to depend on the initial point w to be 0 heavily. Is it possible to allow a random starting point (which doesn't depend on the function)? I don't see an easy fix for it. (Maybe previous proofs for first order methods (like Nemirovsky-Yudin) suffers from the same drawback? If so maybe it's good to clarify)

2. it seems to be possible to extend the lower bound to the query model in the flavor of increment first-order algorithm like in Nemirovsky-Yudin or Agarwal-Bottou? It would be nice if one can allow arbitrary update based on the iterate, gradient, preconditioned gradient.

3. It is mentioned in line 86 that "we conjecture that the optimal algorithm in this case is again .. ". I got confused about why this is a conjecture? Maybe it is good to spell out the gap more explicitly.

Summary: It is a nice paper which provides lower bounds of rounds of communication for a set of interesting functions under very reasonable restriction to the algorithms, which matches DiSCO for quadratic functions. I recommend acceptance.

Submitted by Assigned_Reviewer_3

Light Review: This paper considers the important question of

understanding convex optimization in a distributed setting. The

contribution of the paper is in proving lower bounds on

communication. While this is important, the assumptions on the

algorithms are somewhat restrictive (or looks at the one round case,

which is also I think quite restrictive).

Summary: See below

Submitted by Assigned_Reviewer_4

The paper provides lower bounds for communication required for distributed optimization of separable convex functions.

The key strengths of the paper are: - Very clear presentation of results & contributions in the introduction. - One of the first approaches to look into lower bounds of communication

for distributed convex optimization for both convex and strongly convex

functions. - They show that an accelerated gradient descent method is a method that

achieves the lower bound. - They discuss the case when the separable functions are related to each other. - They present similar bounds for 1-pass algorithms.

The weaknesses of the paper are: - The introduction overstates the results a bit. It should state all the

assumptions made i.e the class of algorithms considered etc.

Quality ======== The quality of the results are very high and the paper is definitely something the larger NIPS community would appreciate.

Clarity ======== The paper very clearly presents the thoughts of the authors. I found very little that needs to be changed. My only suggestion is to be a little more clear (in the introduction) about the assumptions made about the class of algorithms considered.

Originality =========== To the best of my knowledge, the results presented in the paper are original and there isn't any other work that presents similar results.

Significance ============== The paper is significant to the NIPS community.
Summary: The paper very clearly presents lower bounds communication for distributed convex optimization algorithms under certain conditions. I didn't go through the proofs in very much detail but the result look compelling and there is very little wrong with the paper.

Author Feedback
Author rebuttal: We thank all the reviewers for taking the time to study our paper and share their instructive and insightful comments.

Assigned_Reviewer_2
-------------------
Thanks for your comments.
1+2) Indeed, both structural assumptions (initialization point at zero, and new points confined to lie in the span of a certain set) are not strictly required for deriving the type of lower bounds mentioned in the paper. One common way of relaxing these (say, in the papers you cited) is to engineer an orthogonal rotation of the underlying linear space according to the query points issued by the optimization algorithm. For the sake of simplicity, we chose not the delve into these details in the present paper, and present more transparent proofs using slightly stronger assumptions. This choice is relatively standard, and can also be found, for example, in the canonical 'Introductory Lectures on Convex Optimization' by Nesterov. Moreover, our assumptions hold for the vast majority of optimization algorithms designed for this setting, and potentially relaxing them as discussed above does not seem to expand the scope of our lower bounds to other algorithms which are used in practice. We will further clarify these points in the paper.

3) In the non-smooth setting, the distributed subgradient method would require O(1/lambda*epsilon) rounds for lambda-strongly convex functions, and O(1/epsilon^2) for general convex functions, which are larger than the lower bounds we present. That being said, we recently obtained some preliminary results which strongly indicate that this lower bound is indeed tight, using a different algorithm (proximal smoothing combined with distributed accelerated gradient descent). We will add a discussion of this matter in the paper.

Assigned_Reviewer_3
-------------------
Thank you for your comments. We will make sure the assumptions we use are clearly stated.

Assigned_Reviewer_4
-------------------
Thanks for your comments. We would like to emphasize that our work focuses on *lower* bounds which apply for any algorithm satisfying our assumptions. In particular, showing that these lower bounds hold even for simple and natural functions such as quadratics is a strength rather than a weakness of our results.

Assigned_Reviewer_5
-------------------
Thanks for your comments. We would like to emphasize that our work focuses on *lower* bounds which apply for any algorithm satisfying our assumptions, and not to propose specific algorithms for solving the problem at hand or experimental results. Theoretical papers of this type appear in NIPS every year.

Assigned_Reviewer_6
-------------------
Thank you for your kind comments.

Assigned_Reviewer_7
-------------------
Thanks for your comments. We agree that the assumptions pose some restrictions, but as discussed in the paper they hold for a large family of algorithms, including some of the state-of-art methods designed for this setting. Furthermore, it is worth pointing out that the approach taken in the present paper is considerably less restrictive than standard common assumptions which are used in the context of optimization lower bounds, e.g. in 'Introductory Lectures on Convex Optimization' by Nesterov, where it is assumed that points generated by the algorithm lie in the span of previous gradients.